# FinSearchComp: Towards a Realistic, Expert-Level Evaluation of Financial Search and Reasoning

**Liang Hu**[†][*]**, Jianpeng Jiao**[†]**, Jiashuo Liu**[†]**, Dongyuan Mutu**[†]**, Yanle Ren**[†]**, Zhoufutu Wen**[†][*]**,
Kaiyuan Zhang**[†]**, Xuanliang Zhang**[†]
ByteDance Seed
{huliang.will, liniuniu}@bytedance.com

**Xiang Gao**[‡]**, Tianci He**[‡]**, Fei Hu**[‡]**, Yali Liao**[‡]**, Zaiyuan Wang**[‡]**, Jingkai Liu**[‡]**, Daibin Sun**[‡]**,
Ziqing Zeng**[‡]**, Zhiyuan Zeng**[‡]**, Chenghao Yang**[‡]**, Qianyu Yang**[‡]**, Mingren Yin**[‡]**, Xinying Feng**[‡]**,
Ge Zhang**[‡]**, Xinyi Zhang**[‡]**, Xiying Zhao**[‡]**, Zhenwei Zhu**[‡]
ByteDance Seed

**Hongseok Namkoong**
Columbia Business School

**Wenhao Huang**
ByteDance Seed

## Abstract

Search has emerged as core infrastructure for LLM-based agents and is widely viewed as critical on the path toward more general intelligence. Finance is a particularly demanding proving ground: analysts routinely conduct complex, multi-step searches over time-sensitive, domain-specific data, making it ideal for assessing both search proficiency and knowledge-grounded reasoning. Yet no existing open financial datasets evaluate data searching capability of end-to-end agents, largely because constructing realistic, complicated tasks requires deep financial expertise and time-sensitive data is hard to evaluate. We present FinSearchComp, the first fully open-source agent benchmark for realistic, open-domain financial search and reasoning. FinSearchComp comprises three tasks, Time-Sensitive Data Fetching, Simple Historical Lookup, and Complex Historical Investigation, closely reproducing real-world financial analyst workflows. To ensure difficulty and reliability, we engage 70 professional financial experts for annotation and implement a rigorous multi-stage quality-assurance pipeline. The benchmark includes 635 questions spanning global and Greater China markets, and we evaluate 21 models (products) on it. Grok 4 (web) tops the global subset, approaching expert-level accuracy. DouBao (web) leads on the Greater China subset. Experimental analyses show that equipping agents with web search and financial plugins substantially improves results on FinSearchComp, and the country origin of models and tools impact performance significantly. By aligning with realistic analyst tasks and providing end-to-end evaluation, FinSearchComp offers a professional, high-difficulty testbed for complex financial search and reasoning. We provide the data of FinSearchComp in Github and Huggingface.

## 1 Introduction

Search is a core capability for agentic LLMs. Without accurate and timely retrieval, reasoning and decisions are limited. However, the community still lacks realistic benchmarks to assess search in *domain-grounded* and *decision-relevant* settings. General-purpose browsing benchmarks, such as

---

[*]Corresponding authors.

[†]Core contributors, $\alpha$-$\beta$ order.

[‡]Contributors, $\alpha$-$\beta$ order.

Table 1: Comparison of FINSEARCHCOMP with existing financial benchmarks.

| Benchmark | Open-domain Search | Tool Use | Time-sensitive Data | End-to-end Agent Evaluation | Holistic Evaluation |
|---|---|---|---|---|---|
| FinQA (Chen et al., 2021) | ✗ | ✗ | ✗ | ✗ | ✗ |
| ConvFinQA (Chen et al., 2022) | ✗ | ✗ | ✗ | ✗ | ✗ |
| MultiFinBen (Peng et al., 2025) | ✗ | ✗ | ✗ | ✗ | ✗ |
| FinanceQA (Mateega et al., 2025) | ✗ | ✗ | ✗ | ✗ | ✗ |
| BizFinBench (Lu et al., 2025) | ✗ | ✗ | ✗ | ✗ | ✗ |
| FinEval (Guo et al., 2025) | ✗ | ✓ | ✗ | ✓ | ✗ |
| CPA-QKA (Kuang et al., 2025) | ✗ | ✓ | ✗ | ✓ | ✗ |
| Finance Agent (Bigeard et al., 2025) | ✓ | ✓ | ✗ | ✓ | ✗ |
| **FINSEARCHCOMP** (Ours) | ✓ | ✓ | ✓ | ✓ | ✓ |

BrowseComp (Wei et al., 2025), mainly test if agents can complete multi-step navigation to find short, verifiable facts. They avoid long-form synthesis, ambiguity resolution, and do not assess use of domain knowledge or multi-source evidence, limiting for domain-intensive support. In finance, search is central because analysts spend much time searching, validating, and synthesizing information across the web and databases, where timeliness and precision have direct value (Nie et al., 2024; Ding et al., 2024). To address this gap, we study *open-domain financial data search*, where agents retrieve from web and professional databases to answer queries, requiring tool use, time-sensitive data, and integration of heterogeneous evidence (Egg et al., 2025; Ferrag et al., 2025; Peng et al., 2025).

However, existing financial Question-Answering (QA) benchmarks usually pre-collect relevant context and bypass open-domain search and tool use, thereby under-assessing agents' search competence and diverging from analyst workflows Wang et al. (2023); Yang et al. (2024) (see Table 1). Among them, the Finance Agent Benchmark (Bigeard et al., 2025) offers an end-to-end evaluation but is confined to a self-constructed system, which is a base model augmented with a retrieval module. Moreover, the benchmark's use of only historical data permits success via memorization, not necessarily real-time information retrieval.

To address this need, we introduce **FINSEARCHCOMP**, the first *open-domain* benchmark for realistic, analyst-style financial search, comprising 635 questions that require time-sensitive acquisition and multi-source evidence integration. To mirror the day-to-day needs of professional analysts, we design three task families—*Time-Sensitive Data Fetching*, *Simple Historical Lookup*, and *Complex Historical Investigation* (Table 2). These tasks correspond to core analyst workflows. *Time-Sensitive Data Fetching* tasks capture situations where the correct response changes rapidly (e.g., stock prices, exchange rates, and gold prices), emphasizing rapid retrieval and verification under tight time constraints. *Simple Historical Lookup* tasks reflect frequent point-in-time lookups. These questions require going back to the correct historical disclosure and aligning with the company's reporting calendar. *Complex Historical Investigation* tasks involve building multi-period views that integrate different sources. Such queries demand stitching together multiple reports, checking consistency across sources, and ensuring that values are comparable across time.

During construction, we engaged 70 professional financial experts for data annotation, conducted multi-stage verification of prompts and reference answers, and specified detailed, rubric-based scoring guidelines. To account for differences in data sources and reporting conventions, FINSEARCH-COMP covers two subsets: Global and Greater China. Further, since answers of different LLMs may have different formats and exhibit legitimate minor fluctuations (e.g., revisions or rounding across various sources), we adopt an LLM-based evaluation with rubric-guided judging and explicit tolerance bands, complemented by expert spot checks to ensure the overall correctness.

We evaluate 21 models (products) on FINSEARCHCOMP, spanning web-enabled products and API endpoints. On the Global subset Grok 4 (web) attains the highest overall score (68.9%), outperforming the runner-up, GPT-5-Thinking (web), by 5.0 percentage points (pp), yet still trailing human experts by 6.1 pp. On the Greater China subset, Chinese models perform markedly better: DouBao (web) leads the leaderboard, followed closely by YuanBao-DeepSeek-R1 (web) and Grok 4 (web); nevertheless, all remain more than 34 pp below human performance.

Further analyses indicate that equipping agents with web search capabilities and financial plugins improves their performance on FINSEARCHCOMP. Our analysis further identifies recurring failure modes: shallow search, stale or mis-timestamped evidence, cross-unit/currency aggregation, and report-calendar misalignment, offering concrete targets for future improvement. For example, common failures include neglecting to call specialized data plugins in favor of less reliable web searches, extracting incorrect data from a valid source (e.g., confusing opening vs. closing prices), and unnecessarily over-complicating simple queries like "market cap" into multiple steps.

Our contributions are as follows:

1. We introduce FINSEARCHCOMP, the first fully open-source, end-to-end agent benchmark for *open-domain financial data search*. FINSEARCHCOMP comprises 635 expert-curated queries spanning global and Greater China markets and three analyst-style task families (Time-Sensitive, Simple Historical, Complex Historical), with multi-stage quality control.

2. We release a carefully curated benchmark dataset with deterministic gold answers and an fully open-source evaluation harness.

3. We conduct a comprehensive study of 21 models (web-enabled products and APIs), showing that equipping agents with web search and financial plugins consistently improves performance. A case study further reveals that poor performance often stems from insufficient search depth and the retrieval of outdated information.

## 2 FINSEARCHCOMP

We begin by outlining our design principles and choices. We then describe FINSEARCHCOMP 's construction, quality-control measures, and some descriptive statistics.

### 2.1 DESIGN PRINCIPLES

Before introducing FINSEARCHCOMP, we outline the desiderata for a high-quality financial search benchmark and describe how our design addresses them.

1. **Task Professionalism & Diversity.** Financial data retrieval spans tasks of varying complexity, from real-time queries to complex historical analyses. The nuances of financial metrics necessitate domain expert oversight to ensure professional accuracy and relevance.

   ⋆ **Our design:** In consultation with financial analysts, we designed three tasks mirroring their daily workflows: time-sensitive data fetching, simple historical lookup, and complex historical investigation, detailed in Section 2.2.

2. **High Quality of Questions.** Ambiguity arising from varied data sources and definitions can undermine evaluation. Unambiguous questions with clear provenance and reproducible grading criteria are thus crucial for reliability and fairness.

   ⋆ **Our design:** We implement a rigorous quality control process, including reliable data source selection, ambiguity mitigation through precise definitions, and multi-expert answer verification (Section 2.4).

3. **Broad Market Coverage.** To ensure external validity and assess real-world readiness, a benchmark must cover diverse markets, languages, and reporting conventions, revealing generalization failures that single-market tests often miss.

   ⋆ **Our design:** FINSEARCHCOMP includes Global (Western) and Greater China subsets, with questions in English and Chinese. To enable fair comparison, we mirror task templates and balance entity coverage, supporting cross-lingual evaluation (see Section 2.5).

### 2.2 TASK DESIGN OF FINSEARCHCOMP

We define three task types aligned with daily analyst workflows. Each requires at least one external tool call, and each question has a single, fully objective answer. These tasks test core search and reasoning skills that are not only central to financial analysis but also broadly important to knowledge work in many disciplines. Detailed comparison is shown in Table 2.

Table 2: Examples of the three tasks in FINSEARCHCOMP, with retrieval depth, temporal span, reasoning complexity, and typical data types.

| Task | Example | Retrieval Depth | Temporal Span | Reasoning Complexity | Typical Data Types / Examples |
|---|---|---|---|---|---|
| T1. Time Sensitive Data Fetching | *IBM latest close price. Obtained from a real-time query of IBM* | 1 | 1 day | Easy | Stock prices, FX rates, gold prices (real-time quotes) |
| T2. Simple Historical Lookup | *What was the total assets of Starbucks as of September 27, 2020? (Answer: $29374.5 million, rounding errors allowed.)* | 1 | 1 day | Medium | YoY (Year-over-Year), HoH (Half-on-Half), TTM (Trailing Twelve Months), FY (Fiscal Year), quarterly reports |
| T3. Complex Historical Investigation | *From Jan 2010 to Apr 2025, in which month did the S&P 500 index experience the largest single-month increase? (Answer: Apr 2020 (12.68%), error ±0.1% allowed.)* | >1 | 184 months | Hard | Multi-period views, currency/unit normalization, corporate action adjustments, data provenance |

**T1 Time-Sensitive Data Fetching.** Requests intraday/daily data (e.g., close price, filings, guidance), supporting trading, monitoring, and event reactions. Challenges: freshness, calendar alignment, ticker aliasing, conflict resolution. Also relevant to journalism, policy monitoring, or clinical trials.

**T2 Simple Historical Lookup.** Requests fixed-point facts (e.g., FY2024 R&D expense[1], TTM revenue[2]). Enables baselining, YoY[3] / HoH[4] comparisons, event studies, backtests. Challenges: reporting conventions (FY, TTM, quarterly), restatements, unit/currency consistency. Similar needs in policy, medicine, and academia.

**T3 Complex Historical Investigation.** Requests multi-period synthesis (e.g., largest single-month index gain over decades), supporting trend analysis, factor research, valuation comps, risk monitoring. Challenges: long horizons, corporate actions, unit normalization, multi-step reasoning. Also applies to climate science, history, and epidemiology.

## 2.3 CONSTRUCTION OF FINSEARCHCOMP

To accommodate the unique characteristics of different tasks, we employ a variety of data construction strategies to ensure both diversity and quality, as shown in Figure 1 (left side). Example questions of each task can be seen in Section B.4. And we show time cost of financial analysts on three tasks in Section B.3.

**Time-Sensitive Data Fetching.** Financial experts first mannually design questions asking for time-sensitive data that can be verified through API. Time-sensitive data changes over time, so we actually prepare the code for API calls that obtain real-time data points for each question. Finally, financial experts check each API result against the real-time data to ensure the retrieved result is correct and on-time, for accurate evaluation. We establish permissible error margins for evaluation based on the specific volatility characteristics of different assets (e.g., equities, forex).

**Simple Historical Lookup.** We collect data for this task from two sources to enrich the diversity. (i) Financial experts select documents from official sources (e.g., listed company filings, regulatory authority websites, statistics bureau) and extract key data points to formulate questions and their corresponding answers. (ii) We also design questions using historical data with consistent definitions from professional financial databases. To mitigate the risk of data revisions for certain official statistics (e.g., macroeconomic indicators), we address potential ambiguity by setting a reasonable answer range or by explicitly specifying the reference time point in the question.

**Complex Historical Investigation.** The construction process involves two primary methods. (i) Financial experts design questions based on their real-world professional scenarios and get answers

---

[1]Research and development spending in financial statements. FY2024 means fiscal year 2024.

[2]Trailing Twelve Months revenue, sum over most recent 12 months.

[3]Year-over-Year, compares with same period prior year.

[4]Half-over-Half, compares with prior half-year.

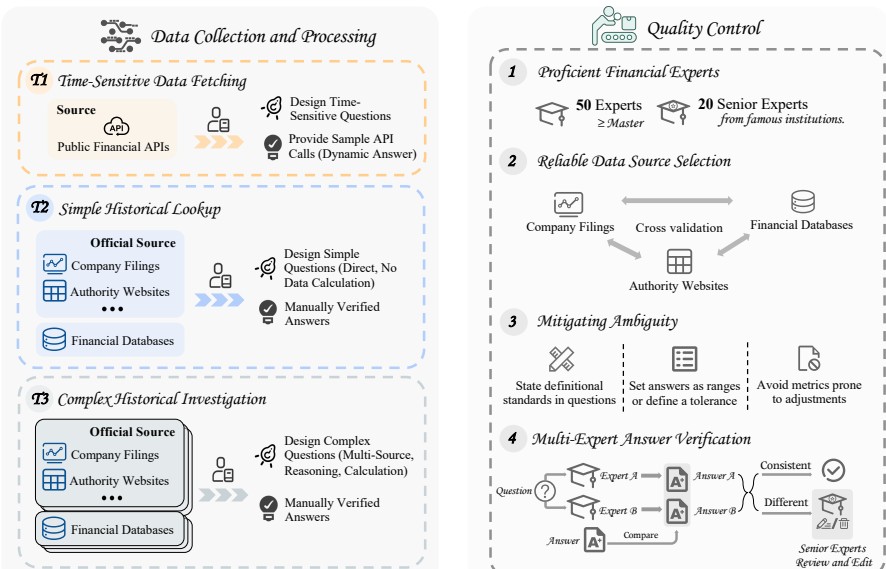

Figure 1: The overview of the construction process. The construction of this benchmark involves three tasks. The data for each task originate from different sources and undergo separate processing pipelines. A uniform quality control procedure is applied across all tasks.

using reliable financial data sources. (ii) Financial experts download tables from a verified and reliable financial database and annotate questions based on the tables. Furthermore, financial experts screen and refine these candidates, finalizing 2 to 5 high-quality questions on each table and recording their definitive answers.

## 2.4 QUALITY CONTROL

To ensure quality, we implemented a rigorous quality-control process throughout construction and summarize the key measures below.

**Proficient Financial Experts.** Our annotation process is managed by financial experts, organized into a 50-member annotation panel and a 20-member senior review panel for arbitration. All experts hold advanced finance degrees and possess active industry experience from leading institutions such as Citadel, J.P. Morgan, and CITIC Securities. Panelists are selected through a rigorous domain-specific assessment to ensure high annotation quality and inter-annotator reliability.

**Reliable Data Source Selection.** To ensure data authority and accuracy, answers are sourced from highly reliable channels, including official company filings, government websites, and professional financial databases. We employ a multi-source cross-validation method to verify data and eliminate ambiguity. This rigorous process allows us to identify and avoid inconsistencies or calculation errors, even those present in widely recognized professional databases.

**Mitigating Ambiguity.** We implement several strategies to mitigate ambiguity. First, we avoid questions involving metrics with inconsistent calculation standards across institutions, such as forward-adjusted stock prices (see Section B.2). Second, we ensure question clarity by explicitly stating definitional standards (e.g., Static PE vs. PE TTM), setting answers as numerical ranges to accommodate minor calculation discrepancies, and avoiding metrics prone to retrospective data revisions where possible. A detailed guide is summarized in Table 4.

**Multi-Expert Answer Verification.** Our verification protocol utilizes a blind review mechanism to ensure accuracy. After an expert drafts a question and answer, one or two other experts solve the question independently without access to the solution. A senior expert arbitrates any discrepancies or ambiguities, leading to the modification or rejection of the question. The entire dataset construction involved approximately 240 expert-hours, ensuring comprehensive professional oversight.

## 2.5 DATA OF FINSEARCHCOMP

We summarize the basic statistics in Figure 6. Most evaluation metrics in FINSEARCHCOMP rely on problem-specific, expert-annotated ranges of acceptable values, rather than a single ground truth. Furthermore, we illustrate the distribution of topics in Figure 7, which covers 10 distinct topics, demonstrating the diversity of FINSEARCHCOMP.

## 2.6 EVALUATION OF FINSEARCHCOMP

**Evaluation Protocol.** Considering the dynamic nature of answers and the need for numerical tolerance in FINSEARCHCOMP, we adopt LLM-as-a-Judge (Zheng et al., 2023) for evaluation. The specific evaluation methods for the three tasks are detailed below, with the corresponding prompts provided in the Section D.

For *Time-Sensitive Data Fetching*, we address several challenges: (*i*) time lags between the model response and the evaluation, (*ii*) potential data latency from some financial APIs, and (*iii*) the inability of most APIs to query prices at a specific second. To mitigate these time-sensitivity issues, we initiate the evaluation process uniformly after the relevant markets have closed. To ensure fairness and accuracy, we establish differentiated evaluation rubrics based on the characteristics of various asset classes, with details in Section C.1. For *Simple Historical Lookup* and *Complex Historical Investigation*, which feature static and deterministic answers, we also annotate rubrics, such as a predefined error margin.

**Evaluation Metrics.** In FINSEARCHCOMP, we use the 0-1 error metric. Ground truth answers are static, except for the Time-Sensitive Data Fetching task, which requires real-time API execution. We employ an LLM judge to assess each candidate answer $A$ against predefined rubrics $R$. This evaluation is modeled by a judgment function, $\mathcal{J}(A, R)$, which returns 1 for a correct answer and 0 otherwise. The final score $S$ is thus defined as:

$$S(A, R) = \mathcal{J}(A, R). \tag{1}$$

**Evaluation Accuracy.** To validate the reliability of LLM-as-a-Judge, we conduct a human evaluation on a representative subset. For each benchmark dataset, we selected 5 models and verified their complete evaluation sets, resulting in roughly 400 instances per dataset. On this sample, the judgments of LLM-as-a-Judge reached 95% agreement with human-verified labels, confirming the robustness of our evaluation protocol. Detailed results are summarized in Table 3.

Table 3: Accuracy of LLM evaluation compared with human evaluation.

| Task | Subset | Accuracy (%) |
|------|--------|--------------|
| T1 | Global | 91.5 |
|    | Greater China | 91.7 |
| T2 | Global | 96.8 |
|    | Greater China | 95.5 |
| T3 | Global | 97.4 |
|    | Greater China | 99.8 |

## 3 EXPERIMENTS

In this section, we present the main results for FINSEARCHCOMP. We evaluate the performance of 21 mainstream models (products) on FINSEARCHCOMP from August $1^{st}$ to $20^{th}$, and to ensure temporal comparability across models, all T1 evaluations were conducted after the official market close (local market time) on each evaluation day. These models are categorized into two groups:

- **Web-based products (12 models):** Grok 4 (web) (xAI, 2025), GPT-5-Thinking (web) (abbr. GPT-5-T (web)), Gemini-2.5-pro (web) (Google, 2025) (abbr. Gemini (web)), Qwen3-235B-A22B-2507 (web) (abbr. Qwen3 (web)) (Qwen Team, 2025), DeepSeek-R1 (web) (DeepSeek-AI et al.), DouBao (web) (ByteDance, 2025), DouBao-Thinking (web) (abbr. DouBao-T (web)), YuanBao-HunYuan-T1-Thinking (web) (abbr. HunYuan-T1 (web)) (Tencent, 2025), YuanBao-DeepSeek-V3 (web) (abbr. YuanBao-V3 (web)), YuanBao-DeepSeek-R1 (web) (abbr. YuanBao-R1 (web)), Ernie-X1 (web) (Baidu, 2025), and Kimi k2 (web) (MoonShot_AI, 2025).

- **APIs (9 models):** Gemini2.5-pro (API) (abbr. Gemini (API)), Gemini-2.5-pro (Google Search) (API) (abbr. Gemini-G (API)), DouBao (API), DouBao-Thinking (API) (abbr. DouBao-T (API)), Qwen3-235B-A22B-2507 (API) (abbr. Qwen3 (API)), DeepSeek-R1 (API), HunYuan-T1 (API), Ernie-X1 (API), and Kimi k2 (API).

To establish a human performance baseline, we engage another 50 financial experts who are not involved in the data construction. These experts perform the benchmark tasks while utilizing search tools. Their average score is taken as the human baseline.

## 3.1 MAIN RESULTS

We report the overall performance of the strongest model from each provider in Figure 8, with the whole performance in Section C. The models form a clear performance hierarchy, with a sizable gap to human experts remaining. Rankings differ between the global and Greater China subsets, likely reflecting differences in training-corpus coverage, language/domain alignment, and retrieval infrastructure. On the global subset, *Grok-4 (web)* and *GPT-5-Thinking* form a clear leading tier, with Grok-4 (web) securing the top score and approaching expert-level accuracy. While Gemini-2.5-pro (web) decline when moving from the global to the Greater China subset, *Grok 4 (web)* remains competitive. On the Greater China subset, *DouBao (web)* and *YuanBao-HunYuan-T1-Thinking (web)* are strong on the Greater China subset, though they still trail human experts by a substantial margin. Detailed results and analyses are shown in Section 3.2 and Section 4.

## 3.2 RESULTS ACROSS DIFFERENT TASKS

We show the results of each tasks respectively in Figure 2. Our main findings are as follows.

**Finding 1. Task difficulty increases from T1 to T3.** We observe a monotonic performance decline from T1 to T3 across all models, confirming the escalating difficulty of our tasks. The complexity of T3 stems from its requirements for multi-hop retrieval, temporal reasoning (e.g., event dating, fiscal-calendar alignment), and fine-grained entity resolution.

**Finding 2. Grok-4 and GPT-5-Thinking approach expert-level reasoning.** While most models trail human performance, Grok-4 and GPT-5-Thinking achieve near-expert results on the global subset. Their performance advantage grows substantially with task difficulty, peaking on T3. This suggests their superiority stems from advanced capabilities in multi-step reasoning, temporal alignment, and entity disambiguation, rather than retrieval alone. Grok-4's top score on T3 for the Greater China subset further corroborates its advanced reasoning capabilities.

## 4 CASE STUDY

In this section, we conduct case studies to analyze the performances in detail. We also provide discussion on the effect of reasoning capabilities in Section E.

## 4.1 HOW MUCH DO SEARCH CAPABILITIES IMPACT PERFORMANCE ON FINSEARCHCOMP?

As shown in Figure 3, models without search uniformly score 0 on T1, as they cannot retrieve current financial data. Without search, they still obtain non-zero but low scores on T2 and T3; we attribute this to parametric memory from pre-training (e.g., annual reports and statistical-agency releases), which surfaces approximate facts that are often outdated or misaligned, yielding higher error rates. With search enabled, average gains of 40.8, 29.0, and 8.1 points are observed on T1, T2, and T3, respectively—largest for time-sensitive tasks but still material for complex historical investigations.

These patterns indicate that FINSEARCHCOMP stresses complex search and reasoning, where success requires planning multi-step queries, aligning timelines and identifiers across sources, and resolving conflicting evidence. In turn, performance reflects not only access to documents but also the ability to verify, synthesize, and reason.

## 4.2 HOW DO FINANCIAL PLUGINS IMPROVE THE PERFORMANCE ON TOP OF GENERAL SEARCH TOOLS?

A comparative analysis reveals that Deepseek R1 on the YuanBao platform, augmented with a financial plugin, significantly outperforms its counterpart on the DeepSeek web interface for financial tasks, as shown in Figure 4. This enhancement is particularly evident in T1, with a 31.9 percentage point improvement. The plugin provides direct access to real-time and historical financial data,

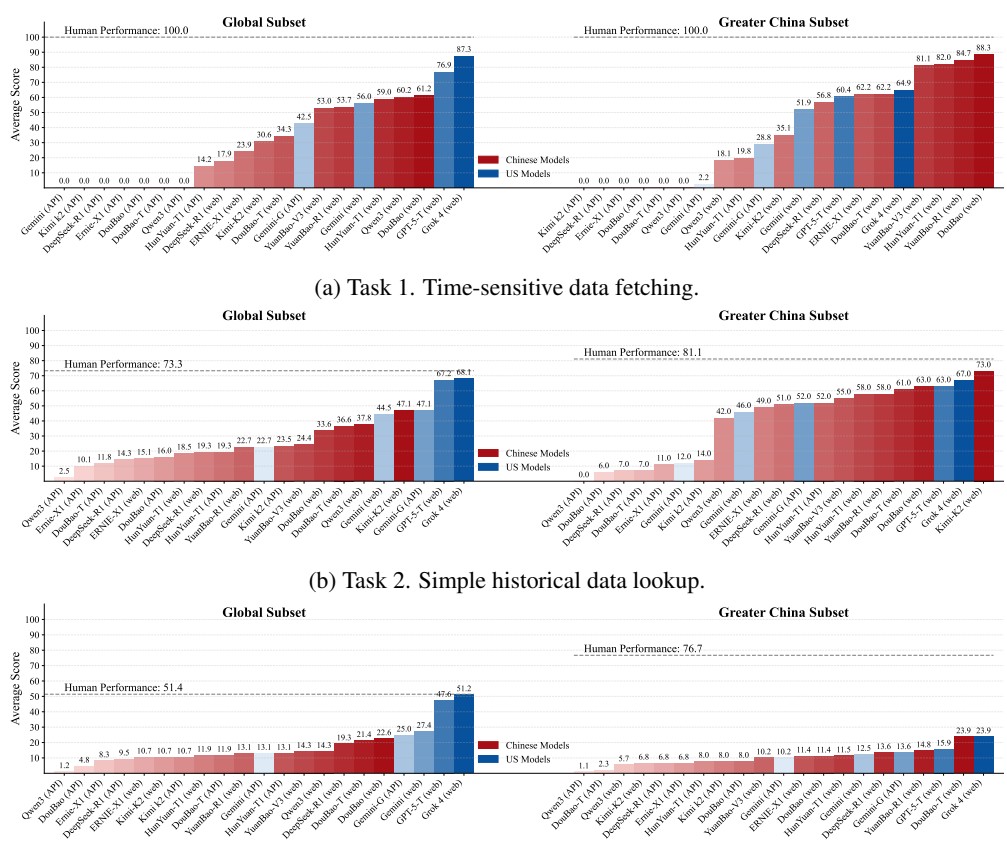

(a) Task 1. Time-sensitive data fetching.

(b) Task 2. Simple historical data lookup.

(c) Task 3. Complex historical investigation

Figure 2: The performance of various models across the three tasks on FINSEARCHCOMP. Models with 0 scores are all APIs.

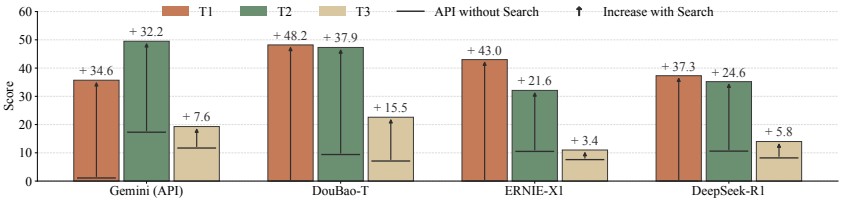

Figure 3: Performance improvement of search-augmented models over models without search capabilities.

such as asset prices and corporate income statements, overcoming the limitations of standard web searches which often yield outdated information. Despite these gains, the performance of YuanBao-R1 (web) remains suboptimal. This suggests that while specialized tools are beneficial, the model's intrinsic capabilities are also critical for effectively searching and reasoning with financial data.

### 4.3 HOW DOES MODEL ORIGIN IMPACT THE PERFORMANCE?

We compare the scores of the US models and the Chinese models on the Global and Greater China subsets in Figure 5. We define asset origin ratio = Global subset scores / Greater China subset scores. Higher ratio means better performance on tasks related to global assets than tasks related to Chinese assets, and vice versa. We observe that models from US tend to show stronger data search performance for global assets, while models of Chinese origin appear more proficient with Chinese assets. This pattern suggests underlying differences in search tool integration or core model capabilities between the two groups. Specially, for tasks T1 and T2, US models consistently yield an asset origin ratio exceeding 100%. In contrast, Chinese models register a ratio below 100%. However, for

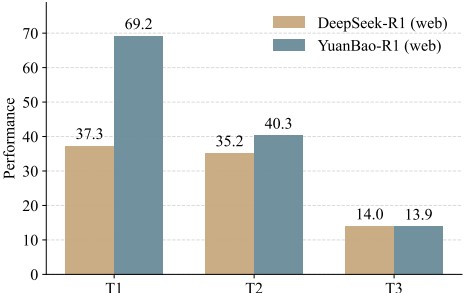

Figure 4: Average performance change on DeepSeek R1 induced by financial plugins.

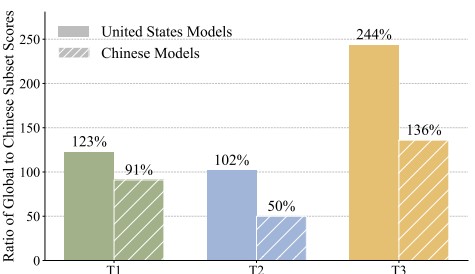

Figure 5: A comparison of US and Chinese models on the ratio of Global to Chinese scores.

task T3, a majority of models achieve a ratio greater than $100\%$. This implies that the global asset challenges within T3 are less demanding than the Chinese asset challenges. Additionally, among the Chinese models, Doubao and Kimi k2 achieve the highest asset origin ratios. The rank suggests their capabilities are more balanced and less skewed toward domestic assets when compared to other models from the same region.

### 4.4 WHERE DO MODELS EXCEL AND FALTER ON FINSEARCHCOMP?

**T1. Time-Sensitive Data Fetching.** Financial plugin–augmented products (GPT-5-Thinking, HunYuan-T1, DouBao; all web) exceed $70\%$ accuracy, outperforming base LLMs. See Section F for details. Deepseek-R1 (web) on its official site scored $28.8$, $12.4\%$ lower than with YuanBao, highlighting the value of specialized retrieval. Failures arise from non-activation, outdated content, or misselection among conflicting sources.

**T2. Simple Historical Lookup.** Grok 4 (web) ranks highest via diverse reliable sources. Some products rely on parametric memory without search, yielding inaccuracies. Most extract from news rather than filings, missing details like prepaid expense.

**T3. Complex Historical Investigation.** Only Grok 4 (web) and GPT-5-Thinking (web) surpass 30. The challenge is structured retrieval (API/SQL), lacking in search-only systems. Successes appear only in simple cases (e.g., difference of two weekly closing prices).

## 5 RELATED WORK

**Financial Benchmarks.** Early efforts such as FINQA (Chen et al., 2021) and its conversational extension CONVFINQA (Chen et al., 2022) target numerical reasoning over annual reports by requiring models to compose multi-step programs that combine text and tabular evidence. Subsequent suites widened both task type and language coverage: FLUE (Shah et al., 2022) aggregated classic tasks in finance, while FINEVAL (Guo et al., 2025) and MULTIFINBEN (Peng et al., 2025) introduced large-scale Chinese and multilingual collections spanning classification, extraction, generation and multimodality. Moving closer to real-world practice, FINANCEQA (Mateega et al., 2025) gathered zero-tolerance questions written by buy-side professionals, and BIZFINBENCH (Lu et al., 2025) distilled $6.7K$ genuine queries from a popular investment-search app to probe long-context and noisy scenarios.

However, these datasets provide relevant financial data by default, which substantially mitigates the challenge of financial data search from open-domain sources (Wei et al., 2025). While Finance Agent Benchmark (Bigeard et al., 2025) incorporates open-domain financial data search, it is limited to searching static historical data. This design introduces the possibility of data memorization by models, thus failing to adequately evaluate their financial data search capabilities.

**Agentic Benchmarks.** To assess end-to-end decision-making, several works frame evaluation as goal-directed interaction with external tools. In finance, the FINEVAL (Guo et al., 2025) agentic track scores models on planning, API use and long-horizon reasoning across tasks such as financial question and answering, financial text classification . Beyond finance, BrowseComp (Wei et al., 2025), BrowseComp-ZH (Zhou et al., 2025), and BrowseComp-Plus (Chen et al., 2025) pose ques-

tions that require persistent web navigation and creative search strategies, offering a challenging yardstick for browsing agents. These studies highlight the gap between token-level metrics and practical autonomy, motivating an evaluation that couples financial expertise with realistic tool use.

## 6 CONCLUSION

In this paper, we address the critical lack of an end-to-end benchmark for evaluating LLM-based agents in financial data search, as prior work failed to assess agent capabilities in realistic, context-free scenarios. To fill this gap, we introduce FINSEARCHCOMP, the first fully publicly available benchmark designed for this purpose. It comprises 635 questions curated by experts across three demanding tasks that require agents to orchestrate various tools, such as SQL, APIs, and web search, to procure verifiable answers. Our holistic evaluation reveals that even state-of-the-art agents significantly underperform humans, often failing due to insufficient search depth and the use of outdated information. We release FINSEARCHCOMP as a vital resource to drive the development of more robust and reliable financial agents.

## 7 REPRODUCIBILITY

We have provided the prompts of this paper in Appendix D. We provide the data of FINSEARCH-COMP in Github and Huggingface.

## 8 ETHICS

All models (products) used in this paper are publicly available, and our usage follows their licenses and terms. Additionally, we confirm that the compensation provided to annotators is significantly higher than the local minimum wage.

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

## A    THE USE OF LARGE LANGUAGE MODELS (LLMS)

We used LLMs only to polish our paper for better clarity and fluency, without involving the core research content. All contents were checked and edited by the authors to ensure the quality and alignment. The authors take full responsibility for the final version of the paper.

## B    DETAILS OF FINSEARCHCOMP

### B.1    RELEASE FORMAT

The dataset is distributed as a set of JSONL files containing {ID, TIER, QUESTION, TOOL_TEMPLATE, ANSWER, TRACE} plus an evaluation harness that replays traces in a sandboxed environment. Detailed documentation and citation files accompany the release.

### B.2    ILLUSTRATION OF INCONSISTENT CALCULATION METHODS FOR THE SAME METRIC

- **Stock Price Adjustment:** Due to significant discrepancies in the calculation of forward-adjusted and backward-adjusted prices across different databases, we uniformly query for non-adjusted prices only.

- **PE (TTM):** The definition of "Earnings" can vary among different institutions.

- **Market Capitalization of Dual-Listed Companies:** We either specify the calculation method (e.g., "Price 1 x Share Class 1 + Price 2 x Share Class 2" vs. "Price 1 x Total Shares") or avoid such questions.

- **Futures Contracts:** The timing for switching the main contract and the algorithm for constructing continuous contracts differ across institutions.

- **Cryptocurrency:** Prices vary across different exchanges.

### B.3    TIME COST OF FINANCIAL ANALYSTS

Financial analysts worldwide dedicate substantial resources to these core information retrieval activities. There are approximately 370,000 financial professionals in the US (based on Bureau of Labor Statistics) and probably over 1 million globally—including equity researchers, portfolio managers, risk analysts, and investment bankers—who regularly perform these tasks as part of their daily workflow.

For T1 (Time-Sensitive Data Fetching), analysts typically spend around 1-2 minutes per query for quick references. This task is also commonly performed by non-expert investors in scenarios such as checking the current price of their investments.

For T2 (Simple Historical Lookup) represents the most frequent task type, with individual analysts performing 10-30 such queries daily for financial analysis, peer comparisons, and modeling. Each lookup averages 5-10 minutes, accounting for data validation (faster for standard financials that can be retrieved from filings, slower for less common financial data such as operational, macroeconomic, and industry data).

For T3 (Complex Historical Investigation) demands the highest time investment per-query, often requiring 15-60 minutes for comprehensive data retrieval and calculation, based on the complexity of the data retrieval and calculation steps. While less frequent (fewer than 20 queries per analyst monthly), these investigations and calculations are important building blocks in financial analysis and reports.

While standardized templates and automated tools already exist to facilitate these tasks—such as comparable company analysis frameworks that can be efficiently updated—approximately half of these information retrieval activities still inevitably require manual data collection and custom analytical framework development. If models could accurately perform such tasks, analysts could further automate these processes and significantly enhance overall productivity.

### B.4 CASES OF THREE TASKS

**For Time-Sensitive Data Fetching.**  Some of the example questions include:

- "Latest closing price of Bloom Energy(NYSE)"
- "The latest opening price of Starbucks"
- "The latest percentage change of Rivian (NASDAQ), based on the latest closing price and the previous closing price"
- "USD/THB price today"
- "Latest price of S&P 500"

**For Simple Historical Lookup.**  Some of the example questions include:

- "What was the additional paid-in capital of Lands' End as of the end of the fiscal year 2020? (answer in thousand dollars, rounded to the nearest integer)"
- "What was the closing value of the VIX on April 25, 2022? (rounded to two decimal places)"
- "For the fiscal year 2023, what was Planet Labs' Net cash provided by investing activities? (please answer in thousands of dollars, rounded to the nearest integer)"
- "What was the U.S. Housing Market Index (HMI) in November 2014? (answer rounded to the nearest integer)"
- "In April 2015, what was the CPI of Russia? (base year 2015=100, rounded to two decimal places)"

**For Complex Historical Investigation.**  Some of the example questions include:

- "What were the specific dates from January 1, 2020, to December 31, 2024, when London Gold (XAUUSD) dropped by more than $80 in a single day? Please list these dates and the corresponding daily drop in USD (rounded to the nearest integer), presented in a table sorted by date in ascending order."
- "During April 2025, did the daily changes (compared to the previous day) in the central parity rates of EUR/CNY, HKD/CNY, and USD/CNY always occur in the same direction (i.e., all rates increasing together or all rates decreasing together)? Among the days when these rates did not move in unison, find the exact date when the USD/CNY central parity rate experienced its greatest single increase. List the three central parity rates (EUR/CNY, HKD/CNY, USD/CNY) for that specific day, rounded to four decimal places."
- "Which constituent stock of the Nasdaq 100 Index (NDX), with a Price-to-Earnings Trailing Twelve Months (PE-TTM) greater than 0 and less than 20 as of the market close on September 30, 2024, reported the largest operating revenue (consolidated financial statements, in billions of USD) in its 2024 interim report? Please provide the stock name and its operating revenue. Unit: billion USD, rounded to two decimal places."
- "What is the year-over-year change in the proportion of Johnson & Johnson's revenue from international markets (excluding the United States) for each of the past three years (2022–2024)? Please provide the change in percentage points, rounded to two decimal places."

### B.5 GUIDE FOR MITIGATING AMBIGUITY

Table 4: Consolidated guide for annotation in FINSEARCHCOMP for mitigating ambiguity.

| Category | Topic | Description | Bad Example | Good Example |
|---|---|---|---|---|
| **Corporate Fundamentals** | Calendar vs. Fiscal Year | Questions must differentiate between calendar and fiscal years. Many companies (e.g., NVIDIA) do not align their fiscal year with the calendar year. Default to "fiscal year" for consistency. | "What was NVIDIA's revenue in 2024?" | "What was NVIDIA's revenue for fiscal year 2024?" |
| | Timing Description for Financial Statements | Use precise language for time periods. Income statement and cash flow items occur "over a period", while balance sheet items are a snapshot "at a point in time". | "What were the company's assets in fiscal year 2023?" | "What were the company's total assets as of the end of fiscal year 2023?" |
| | Financial Item Naming | The variable name in the question must match the terminology used in the financial statements to avoid ambiguity (e.g., "operating income" vs. "operating revenue"). | "What was the company's operating revenue?" (when the report lists "operating income") | "What was the company's operating income?" |
| **Market Data** | GAAP vs. Non-GAAP | To prevent evaluation mismatches, questions must specify the standard (GAAP or Non-GAAP). This ensures the ground truth and the answer are based on the same accounting principles. | "What was the company's income?" (Ambiguous; the ground truth might be GAAP while the answer is Non-GAAP) | "What was the company's net income, based on U.S. GAAP standards?" |
| | Currency | To prevent evaluation mismatches, questions must specify the currency (e.g., USD, CNY). This ensures the answer can be directly compared to the ground truth. | "What was the company's revenue?" | "What was the company's revenue in millions of USD?" |
| | Industry Classification | If a company's industry is mentioned, specify the classification standard (e.g., a specific level of Shenwan or CSRC industry codes) to ensure consistency. | "What industry is the company in?" | "What is the company's industry classification according to the Shenwan Level 1 standard?" |

*Continued on next page*

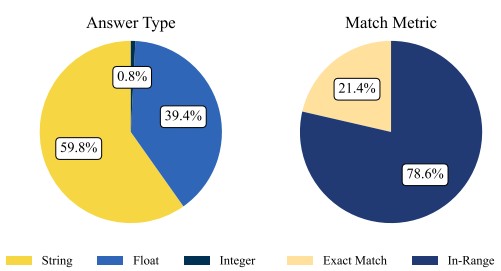 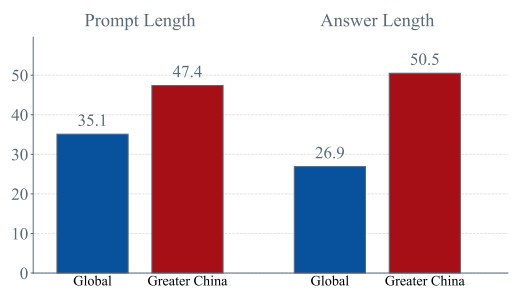

(a) Distribution of the answer type and evaluation rubrics.

(b) Average length of prompts and answers on each subset.

Figure 6: Data statistics of FINSEARCHCOMP.

**Table 4 – Continued from previous page**

| Category | Topic | Description | Bad Example | Good Example |
|---|---|---|---|---|
| | Market Capitalization | For multi-listed companies, specify the exact calculation method. A simple "Market Cap = Total Shares" is ambiguous; a sum of market values from each listing is precise. | "What is the total market cap of a company dual-listed in Hong Kong and Shanghai?" | "What is the total market cap of the dual-listed company, calculated as (A-share price × A-share count) + (H-share price × H-share count)?" |
| **Fixed Income & Macro** | Futures Quote Notation | Futures quotes can use special hexadecimal notation (e.g., 113'08'5). The answer should accept both this format and the standard decimal equivalent to be robust. | Answer requires "113.265625" only, but "113'08'5" is also acceptable. | The reference answer is "113.265625, but 113'08'5 is also acceptable." |
| | Currency Exchange Rates | Specify the type of RMB exchange rate: onshore (CNY), offshore (CNH), or interbank, as their values differ. | "What is the USD to RMB exchange rate?" | "What is the onshore USD to CNY exchange rate as of [Date]?" |
| **General Rules** | Answer Precision | Questions must specify the required precision for numerical answers (e.g., number of decimal places, rounding to nearest integer). | "What is the profit margin?" | "What is the profit margin in percentage, rounded to two decimal places?" |
| | Unit Specification | Clearly state the unit for the answer (e.g., million, billion, USD, %). | "What was the revenue?" | "What was the revenue in billions of USD, rounded to the nearest integer?" |

### B.6 DATA STATISTICS

## C DETAILED SCORES ON FINSEARCHCOMP

We show the detailed scores of various models on FINSEARCHCOMP in Table 5.

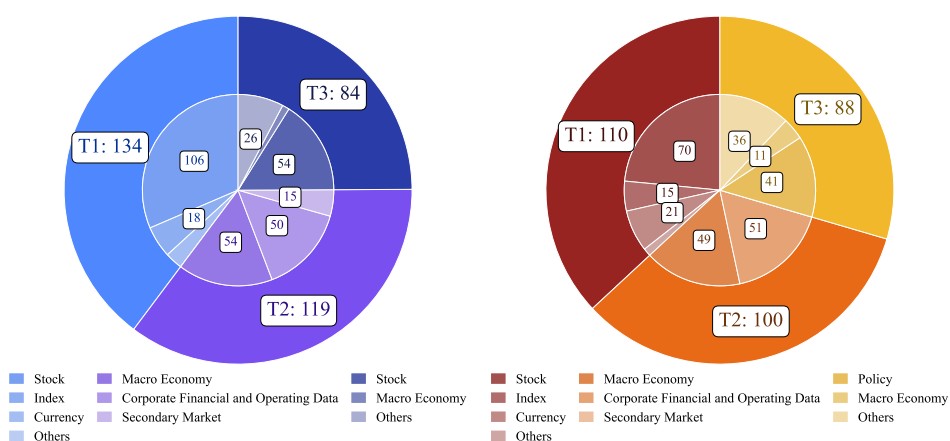

(a) Topic distribution of Global subset.      (b) Topic distribution of Greater China subset.

Figure 7: Topic distributions in FINSEARCHCOMP.

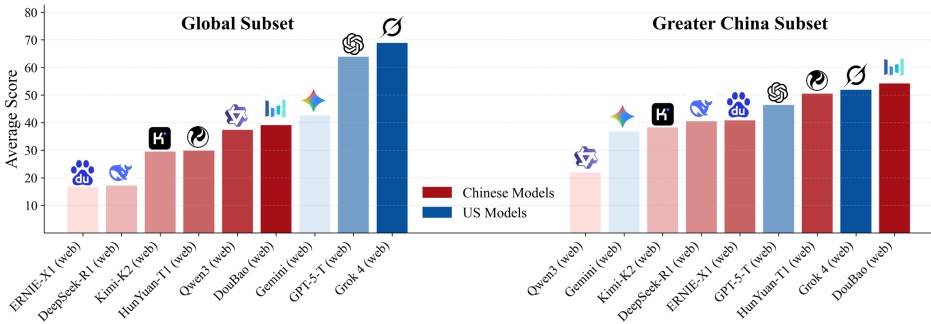

Figure 8: The performance of web-based products on the global subset (left) and the Greater China subset (right) of FINSEARCHCOMP. Note that the performance of human experts is 75.0 and 88.3 on the Global and Greater China subsets, respectively.

## C.1 RUBRICS WITH VARIOUS ASSET CLASSES

- Mainstream Market Stocks and Indices: Evaluation is conducted during non-trading hours. Only minor discrepancies attributable to rounding are permitted.
- Other Regional Indices: An answer is considered correct if its value falls within the day's high-low price range.
- Foreign Exchange (FX) Rates: To account for potential discrepancies across different data providers, the valid range for an answer is defined as the high-low range of the day, augmented by an additional buffer.

## D PROMPT

This section primarily describes the judge system prompts on three subtasks.

> **Judge for Time-Sensitive Data Retrieval**
>
> You are a strict judge. Your task is to score a student's response to a financial question based on the question itself, the Real-time Authentic Information I provide, and the Scoring Criteria. A score of 1 means the student's response meets the requirements, and 0 means it does not. Please provide your analysis first, then give the final score. If the final score is 1, output '{"score":1}'; if it is 0, output '{"score":0}'. You must output strict JSON.
>
> The specific rules are as follows:

Table 5: Performance of various models and human on FINSEARCHCOMP.

| Model | Reasoning | Search | Global | | | | Greater China | | | | Avg. |
|---|---|---|---|---|---|---|---|---|---|---|---|
| | | | T1 | T2 | T3 | Avg. | T1 | T2 | T3 | Avg. | |
| Human Performance | - | ✓ | 100.0 | 73.3 | 51.4 | 75.0 | 100.0 | 88.1 | 76.7 | 88.3 | 81.6 |
| *Web-based products* | | | | | | | | | | | |
| Grok 4 | ✓ | ✓ | 87.3 | 68.1 | 51.2 | 68.9 | 64.9 | 67.0 | 23.9 | 51.9 | 60.4 |
| GPT-5-Thinking | ✓ | ✓ | 76.9 | 67.2 | 47.6 | 63.9 | 60.4 | 63.0 | 15.9 | 46.4 | 55.2 |
| Gemini2.5-pro | ✓ | ✓ | 56.0 | 44.5 | 27.4 | 42.6 | 51.9 | 46.0 | 12.5 | 36.8 | 39.7 |
| DouBao | | ✓ | 61.2 | 33.6 | 22.6 | 39.1 | 88.3 | 63.0 | 11.4 | 54.2 | 46.7 |
| DouBao-Thinking | ✓ | ✓ | 34.3 | 33.6 | 21.4 | 29.8 | 62.2 | 61.0 | 23.9 | 49.0 | 39.4 |
| YuanBao-HunYuan-T1-Thinking | ✓ | ✓ | 59.0 | 18.5 | 11.9 | 29.8 | 82.0 | 58.0 | 11.5 | 50.5 | 40.1 |
| YuanBao-DeepSeek-V3 | | ✓ | 53.0 | 24.4 | 14.3 | 30.5 | 81.1 | 55.0 | 10.2 | 48.8 | 39.7 |
| YuanBao-DeepSeek-R1 | ✓ | ✓ | 53.7 | 22.7 | 13.1 | 29.8 | 84.7 | 58.0 | 14.8 | 52.5 | 41.2 |
| Kimi k2 | ✓ | ✓ | 30.6 | 47.1 | 10.7 | 29.5 | 35.1 | 73.0 | 6.8 | 38.3 | 33.9 |
| Qwen3-235B-A22B-2507 | ✓ | ✓ | 60.2 | 37.8 | 14.3 | 37.4 | 18.1 | 42.0 | 5.7 | 21.9 | 29.7 |
| DeepSeek-R1 | ✓ | ✓ | 17.9 | 19.3 | 14.3 | 17.2 | 56.8 | 51.0 | 13.6 | 40.5 | 28.8 |
| ERNIE-X1 | ✓ | ✓ | 23.9 | 15.1 | 10.7 | 16.6 | 62.2 | 49.0 | 11.4 | 40.8 | 28.7 |
| *APIs* | | | | | | | | | | | |
| Gemini2.5-pro (Google Search) | ✓ | ✓ | 42.5 | 47.1 | 25.0 | 38.2 | 28.8 | 52.0 | 13.6 | 31.5 | 34.8 |
| Gemini2.5-pro | ✓ | | 0.0 | 22.7 | 13.1 | 11.9 | 2.2 | 12.0 | 10.2 | 8.1 | 10.0 |
| Hunyuan-T1-latest | ✓ | ✓ | 14.2 | 19.3 | 13.1 | 15.5 | 19.8 | 52.0 | 8.0 | 26.6 | 21.1 |
| Kimi k2 | ✓ | | 0.0 | 23.5 | 10.7 | 11.4 | 0.0 | 14.0 | 8.0 | 7.3 | 9.4 |
| DeepSeek-R1 | ✓ | | 0.0 | 14.3 | 9.5 | 7.9 | 0.0 | 7.0 | 6.8 | 4.6 | 6.3 |
| ERNIE-X1 | ✓ | | 0.0 | 10.1 | 8.3 | 6.1 | 0.0 | 11.0 | 6.8 | 5.9 | 6.0 |
| DouBao | | | 0.0 | 16.0 | 4.8 | 6.9 | 0.0 | 6.0 | 8.0 | 4.7 | 5.8 |
| DouBao-Thinking | ✓ | | 0.0 | 11.8 | 11.9 | 7.9 | 0.0 | 7.0 | 2.3 | 3.1 | 5.5 |
| Qwen3-235B-A22B-2507 | ✓ | | 0.0 | 2.5 | 1.2 | 1.2 | 0.0 | 0.0 | 1.1 | 0.4 | 0.8 |

- If the Student Answer is empty, score 0.
- If the Student Answer is not empty, but the data in the Real-time Authentic Information is empty (contains no numbers), output '{"score":"null"}'.
- The "Scoring Criteria - Required Content" is the minimum information the student must provide. If the student's response does not meet the Required Content (e.g., is missing content), score 0. Providing extra information is not an error and will not result in a deduction. You only need to focus on the information itself, not the time. You should not deduct points because the question asks for the "latest price" and the student only provides the "price on month x, day x."
- The "Real-time Authentic Information" is the **correct response information** for the user's question at the time of inquiry, and you should completely trust it. If the student's response meets the Required Content, use the Real-time Authentic Information as the ground truth to verify the correctness of the student's answer. Score it according to the "Scoring Criteria - Accuracy Requirements": score 1 if it meets the requirements, and 0 if it does not.
- If the "Scoring Criteria - Accuracy Requirements" is "Only rounding errors are allowed," it means a difference between the student's number and the real-time information is permissible if it's caused by rounding to a different number of decimal places. For example, 12.26 vs. 12.256 is considered a match, scoring 1.
- If the "Scoring Criteria - Accuracy Requirements" is "Any answer between the high and low price is considered correct," it means the student's number is correct if it falls between the maximum and minimum values provided in the Real-time Authentic Information. Rounding errors are also allowed. For example, if the Real-time Authentic Information gives a high of 12.256 and a low of 12.152, a student answer of 12.26 is considered within the correct range and scores 1, while an answer of 12.27 is not and scores 0.
- If the "Scoring Criteria - Accuracy Requirements" is "The allowable error range is an absolute value of ±xx," it means the student's number is correct if it falls within the range of the Real-time Authentic Information's number ±xx. For example, if the requirement is "The allowable error range is an absolute value of ±0.01%," and the number in the Real-time Authentic Information is 13.56%, then any student answer within the range [13.55%, 13.57%] is correct and scores 1; otherwise, it scores 0.
- If the "Scoring Criteria - Accuracy Requirements" is "Expand the high-low price range by an absolute value of a," the correct range is from "minimum value - a" to "maximum value + a" based on the Real-time Authentic Information.

Note:
- You only need to verify numerical values like stock prices/indices, not dates. As long as the numerical values are correct, the student's response is considered correct. You should not deduct points if the student's response mentions "latest/yesterday" while the reference information specifies a particular date (e.g., "Month X, Day Y"), nor should you deduct points if the dates do not align (due to potential issues with time zones, trading days vs. calendar days, etc.).

The following examples use fictional dates and numbers. So do not refer to the specific figures.

Example 1:
Question: How is NVDA's stock market performance today?
Scoring Criteria:
Required Content: NVIDIA's latest opening price, high price, low price, and change percentage.
Accuracy Requirements: Only rounding errors are allowed
Real-time Authentic Information:
{
"data": {
"symbol": "NVDA",
"open": "112.905",
"high": "114.665",
"low": "112.660",
"price": "113.820",
"latest_trading_day": "20250426",
"previous_close": "114.500",
"change": "-0.680",
"change_percent": "-0.59%"
}
}
Student Answer:
According to the latest news as of 17:47 Beijing time on April 27, at the close of the most recent U.S. trading day, NVIDIA's (NVDA) stock price fell by 0.59%.

On that day, NVIDIA's opening price was $112.91, the high was $114.67, the low was $112.66, the volume was 130 million, and the turnover was $15.15 billion. The after-hours price was $113.28, down $0.54 from the closing price, a decrease of 0.47%.
Your Output:
Analysis:
Does the student's answer meet the Required Content: Yes;
Are the numbers consistent (allowing for rounding errors): The opening, high, and low prices are consistent, with only rounding errors; The student's answer about the change percentage (-0.47%) is not consistent with the real-time information's "change_percent": "-0.59%".
Not all requirements are met, final score is 0;
Final score: {"score":0}

Example 2:
Question: USD/CNY onshore exchange rate
Scoring Criteria:
Required Content: The latest onshore USD/CNY exchange rate
Accuracy Requirements: Any answer between the high and low price is considered correct
Real-time Authentic Information:
{
"currency_pair": "USD/CNY",
"exchange_per": "-0.0403",
"exchange_range": "-0.0029",
"exchange_rate": "7.1871",
"exchange_rate_hi": "7.1934",
"exchange_rate_lo": "7.1855",
"open_exchange_rate": "7.1900",
"pre_close_exchange_rate": "7.1900",
"trading_date": "20250605"
}

Student Answer:

As of 03:00 Beijing time on June 5, 2025, the onshore yuan (CNY) closed at 7.1905 against the US dollar, down 26 points from the previous trading day's night session close, with a trading volume of $36.094 billion.

Additionally, on June 5, 2025, the central parity rate of the RMB in the inter-bank foreign exchange market was 7.1729 yuan per US dollar, an appreciation of 32 basis points from the previous trading day.

Your Output:

Analysis:

Did the student answer the Required Content: Yes;

Does the number meet the accuracy requirement: The threshold range from the real-time information is [7.1855, 7.1934]. The student's answer, 7.1905, falls within this range and meets the requirement.

Final score: {"score":1}

Example 3:

Question: Latest COMEX gold price

Scoring Criteria:

Required Content: Latest COMEX gold price

Accuracy Requirements: The allowable error range is an absolute value of ±0.6

Real-time Authentic Information:

```
{
"res": {
"request": "w.wsq("GC.CMX","rt_date,rt_time,rt_last")",
"code": 0,
"data": {
"GC.CMX": {
"RT_DATE": 20250619,
"RT_TIME": 90628,
"RT_LAST": 3383.2000
}
}
}
}
```

Student Answer:

According to Hithink RoyalFlush Futures data, as of 23:22 on June 18, 2025, the price of New York gold (COMEX Gold Futures) was $3382.7 / ounce, down $25.4 from the previous trading day, a decrease of 0.75%, with a high of $3405.2 / ounce and a low of $3363.6 / ounce.

Your Output:

Analysis:

Student's answer meets the Required Content: Yes;

Does the number meet the accuracy requirement: The threshold from the real-time information is 3383.2000 ± 0.6, which is [3382.6000, 3383.8000]. The student's answer, 3382.7, falls within this range and meets the requirement.

Final score: {"score":1}

=======End of examples, this is your task=======

## Judge for Single Historical Data Point Retrieval and Complex Data Retrieval

You are an intelligent judge and scorer for answers to financial questions. You will receive a Question, its Reference Answer, and a Student Answer. Some Reference Answers may be supplemented with "Scoring Criteria". You need to evaluate the Student Answer and complete the following tasks:

1. Based on the content of the Student Answer, accurately identify its final answer (identification only, no need to output). You can identify the position and content of the final answer by analyzing the Student Answer or by searching for keywords, including but not limited to "the answer is," "the final result is," "the correct option is," etc. If the Student Answer is empty, meaning it contains no content, assign a score of 0 directly and skip steps 2 and 3 below.

2. Separately list the final answer from the Reference Answer and the final answer you identified from the Student Answer, and compare the two (no need to output the listing and comparison process or results).

3. Based on the result of the comparison and any Scoring Criteria that may be provided with the Reference Answer, judge whether the Student Answer is correct and assign a score. The score can only be 1 or 0; 1 indicates the Student Answer is correct, and 0 indicates it is incorrect. No scores other than 0 and 1 are permitted.

*Notes:**

1. You do not need to and should not answer or solve the question yourself. Your sole task is to judge and score.
2. The Reference Answer is accurate and correct. You can fully trust it.
3. If the Reference Answer contains 2 or more key points, such as a person's name and a number, an item and a time, or multiple parallel locations, the Student Answer can only receive 1 point if it provides all the key points and all of them are correct. If only a portion of the key points are provided or are correct, the score is 0.
4. If the Student Answer is empty or an abnormal error message, please assign a score of 0.
5. Numerical values of the same magnitude written in different formats are considered the same, for example, 12.45% vs 0.1245, or 120,400,000 vs 120.4 million, or 2/5 vs 0.4. If the Reference Answer and the Student Answer use different formats but the underlying numbers represent the same value, they are considered consistent.
6. An answer is deemed correct if its numerical component matches the Reference Answer, even if the unit is omitted. For example, if the Question asks for "xx revenue in billions of yuan", a Student Answer of "12.98" is a valid match for the Reference Answer "12.98 billion yuan". Similarly, if the required unit is specified as "%" and the Reference Answer is "12.98%", a Student Answer of "12.98" is also a valid match.

*Regarding your output format:**
You need to first output the evaluation result, then provide the scoring basis and the score. Your output format is:

- Scoring Basis: (Briefly state your reason for the score, not exceeding 150 words or 150 tokens)
- JSON:
```
{"answer_score": your score}
```

###
*Example 1**
Question:
What is the average number of years it took for the C9 League university presidents who spoke at the 2024 graduation ceremonies to become a president of a university for the first time after receiving their PhD? (Calculate in years, round to one decimal place) Reference Answer:
27.9 years
Scoring Criteria:
The number in the Student Answer must be strictly identical to the number in the Reference Answer.
Student Answer:
Based on known information, the average time for the current presidents of C9 League universities to become a president for the first time after their PhD is approximately 27.3 years. Here is the specific calculation:
1. Peking University: President Gong Qihuang, PhD in 1992, became president in 2022, a gap of 30 years.
2. Tsinghua University: President Wang Xiqin, PhD in 1996, became president in 2022, a gap of 26 years.
3. Fudan University: President Jin Li, PhD in 1994, became president in 2021, a gap of 27 years.
4. Shanghai Jiao Tong University: President Ding Kuiling, PhD in 1990, became president in 2021, a gap of 31 years.
5. Nanjing University: President Lü Jian, PhD in 1993, became president in 2018, a gap of 25 years.
6. University of Science and Technology of China: President Bao Xinhe, PhD in 1991, became president in 2017, a gap of 26 years.
7. Xi'an Jiaotong University: President Wang Shuguo, PhD in 1986, became president in 2014, a gap of 28 years.
8. Zhejiang University: President Wu Zhaohui, PhD in 1993, became president in 2015, a gap of 22 years.

9. Harbin Institute of Technology: President Han Jiecai, PhD in 1991, became president in 2022, a gap of 31 years.
Summing the years:
30 + 26 + 27 + 31 + 25 + 26 + 28 + 22 + 31 = 246
Average time:
246 ÷ 9 = 27.3 years
Therefore, on average, it took these presidents approximately 27.3 years to become a university president for the first time after their PhD.
- Scoring Basis: This question has Scoring Criteria, which require the number in the Student Answer to be strictly identical to the number in the Reference Answer. Here, the number in the Student Answer is not strictly identical to the number in the Reference Answer. Therefore, the Student Answer is incorrect, and the score is 0.
- JSON:
```
{"answer_score": 0}
```

###
*Example 2**
Question:
An interesting aspect of the United States is that a state's political center is often not its economic center. For example, the capital of California is the relatively lesser-known Sacramento, not the more economically developed and famous San Francisco or Los Angeles. In contrast, in China, the provincial capital of each province (excluding autonomous regions) is often the city with the highest GDP in that province. Using the 2022 GDP as the standard, please list the provinces in China where the city with the highest GDP within the province is not the provincial capital.
Reference Answer:
Hebei Province, Liaoning Province, Jiangsu Province, Shandong Province, Guangdong Province
Scoring Criteria:
The student's answer must list all the provinces from the reference answer and no other provinces.
Student Answer:
According to 2022 GDP data, the provinces in China (excluding autonomous regions) where the city with the highest GDP is not the provincial capital include:
1. Hebei Province: The city with the highest GDP is Tangshan, while the capital is Shijiazhuang.
2. Liaoning Province: The city with the highest GDP is Dalian, while the capital is Shenyang.
3. Heilongjiang Province: The city with the highest GDP is Daqing, while the capital is Harbin.
4. Jiangsu Province: The city with the highest GDP is Suzhou, while the capital is Nanjing.
5. Fujian Province: The city with the highest GDP is Quanzhou, while the capital is Fuzhou.
6. Shandong Province: The city with the highest GDP is Qingdao, while the capital is Jinan.
7. Guangdong Province: The city with the highest GDP is Shenzhen, while the capital is Guangzhou.
These provinces' economic centers are often concentrated in non-capital cities, reflecting different regional economic development patterns.
- Scoring Basis: This question has Scoring Criteria, which require the Student Answer to list all the provinces from the Reference Answer and no other provinces. Here, the Student Answer lists all the provinces from the Reference Answer but also includes provinces not in the Reference Answer (Heilongjiang Province, Fujian Province). Therefore, the Student Answer is incorrect, and the score is 0.
- JSON:
```
{"answer_score": 0}
```

###
*Example 3**
Question:
How many Michelin three-star restaurants in San Francisco specialize in Asian cuisine, and what are their names?
Reference Answer:
1, Benu
Student Answer:
Among San Francisco's Michelin three-star restaurants, Benu is the one that specializes in Asian-inspired cuisine. Helmed by renowned chef Corey Lee, Benu offers modern Asian cuisine, fully showcasing Lee's creativity and skill. The dishes blend Korean, Chinese, and other Asian elements

with a high degree of innovation and refinement. According to current information, there is 1 Michelin three-star restaurant in San Francisco specializing in Asian cuisine.
- Scoring Basis: This question does not have Scoring Criteria. Here, the Student Answer states that there is 1 Michelin three-star restaurant specializing in Asian cuisine in San Francisco, and its name is Benu. Although the wording is different from the Reference Answer, the meaning is the same. Therefore, the Student Answer is correct, and the score is 1.
- JSON:
```
{"answer_score": 1}
```

###
*Example 4**
Question:
In terms of area, which provincial capital city in China (mainland provinces) is closest in size to Slovenia? What is the difference in square kilometers?
Reference Answer:
It is closest to Kunming, with a difference of about 740 square kilometers.
Student Answer:
The total area of Slovenia is 20,273 square kilometers. Among China's provincial capitals, the one closest in area is Kunming, with an area of 22,100 square kilometers. The difference between the two is: 22,100 - 20,273 = 1827 square kilometers.
- Scoring Basis: This question does not have Scoring Criteria. Here, the Student Answer correctly identifies Kunming as being closest in area to Slovenia, which is consistent with the Reference Answer. However, the area difference of "1827 square kilometers" given in the Student Answer is inconsistent with "about 740 square kilometers" in the Reference Answer. Therefore, the Student Answer is incorrect, and the score is 0.
- JSON:
```
{"answer_score": 0}
```

###
I hope you are competent for this job, as it is very important to me. If you perform well, I will reward you accordingly. Otherwise, I may penalize you.

# E  ADDITIONAL ANALYSIS EXPERIMENTS

## E.1  DOES REASONING ABILITY ENHANCE PERFORMANCE ON FINSEARCHCOMP?

An evaluation of models within the same series is detailed in Figure 9. We observe an average decline of 7.0 points for T1 for reasoning capacity, which is likely due to the low complexity of the task and potential overthinking of reasoning models (Aggarwal et al., 2025). For T2 and T3, the change for adding reasoning capacity is negligible.

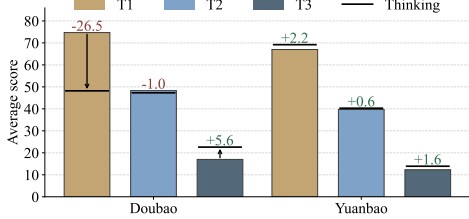

Figure 9: Performance changes from the reasoning capability across models in the same series.

# F  CASE

In this section, we present typical cases identified during evaluation on FINSEARCHCOMP.

A web-based product successfully addresses a question of Time-Sensitive Data Fetching in FIN-SEARCHCOMP by leveraging financial plugins.

**Question**
walmart current stock price

**Response**
As of the U.S. market close on August 27, 2025, the stock price of Walmart Inc. (Ticker: WMT) was $96.08, marking a daily gain of 0.03% with a trading turnover of $1.241 billion. The price represented a slight increase from the previous day's close of $96.05. During the session, the stock reached an intraday high of $96.32 and a low of $95.60.
In terms of recent performance, Walmart's stock has declined by 6.33% over the past five trading days, yet it still maintains a year-to-date (YTD) gain of 7.13%. Analysts note that despite facing short-term pressure from tariff costs, the company has sustained its market competitiveness through supply chain optimization and its omnichannel strategy.

**Search Sources**

1. tradersunion.com
   Walmart Inc. (WMT) Stock Price Forecast for 30 days: Today, Tomorrow, Next Week
   *August 27, 2025 - The current price of Walmart Inc. (WMT) is US$96.02. The analytical forecast indicates that the price of WMT, expected to reach US$96.12 by the end of the week, could rise/fall to US$99.35 by the end of the month.*

2. tradersunion.com
   Walmart Inc. (WMT) Stock Price Forecast for 30 days: Today, Tomorrow, Next Week
   *August 27, 2025 - The current price of Walmart Inc. (WMT) is US$96.02. The analytical forecast indicates that the price of WMT is expected to reach US$96.12 by the end of the week, and could rise/fall to US$99.35 by the end of the month.*

3. in.benzinga.com
   Walmart's Options: A Look at What the Big Money is Thinking - Walmart (NYSE:WMT) - Benzinga
   *August 26, 2025 - Whales with a lot of money to spend have taken a noticeably bearish stance on Walmart. Looking at options history for Walmart WMT we detected 15 trades. If we consider the specifics of each trade, it is accurate to state that 33% of the investors opened trades with bullish expectations and 46% with bearish.*

4. **Tencent Financial Industry Database**
   Walmart WMT.N
   *August 28, 2025 - Real-time Stock Quote for Walmart (Code: WMT.N) in US Stocks: Data Updated: 2025-08-27 16:04:58; Latest Price: $96.08; Previous Close: $96.05; Today's Open: $95.90; Day's High: $96.32; Day's Low: $95.60; Day's Change: 0.03%; Day's Amplitude: 0.74%; Turnover Rate: 0.16%; Volume (shares): 12,925,656; Turnover (USD): 1,241,094,879.00; Volume Ratio: 0.45; 5-Day Change: -6.33%; YTD Change: 7.13%; P/E Ratio (TTM): 36.26; P/B Ratio: 8.51; Dividend Yield: 0.98%; 20-Day Change: -1.32%; Total Market Cap (billion USD): 766.76.*

5. tradersunion.com
   Walmart Inc. (WMT) Stock Price Forecast (30 days): Today, Tomorrow, Next Week
   *August 27, 2025 - The current price of Walmart Inc. (WMT) is $96.02. According to analytical forecasts, the price of WMT is expected to reach $96.12 by the end of this week, and could rise/fall to $99.35 by the end of this month.*

6. marketbeat.com
   Walmart (NYSE:WMT) Price Target Raised to $114.00
   *August 24, 2025 - Walmart (NYSE:WMT) Price Target Raised to $114.00 Key Points - The Goldman Sachs Group has raised Walmart's price target from $101.00 to $114.00, indicating a potential upside of 17.25% from its previous close.*

7. Tonghuashun Finance
   Walmart's turnover on August 27 was $1.241 billion, a decrease of 41.91% from the previous trading day.
   *August 28, 2025 - On August 27, 2025, consumer staples retailer Walmart (WMT) had a turnover of $1.241 billion, ranking 44th in US stocks for the day. The turnover decreased by 41.91% from the previous day, and the trading volume was 12.9257 million shares.*

8. WeChat Official Account · CC Finance
   Walmart, with Unimpressive Revenue and Profits, Has a Stock Price Stronger Than the

Magnificent Seven
*July 29, 2025 - The continuous rise of Walmart's stock price (around $95.67/share in July 2025, market cap of $760 billion, up over 300% from its 2020 low), despite single-digit growth in revenue and profit...*

9. WeChat Official Account · Haishi Shibei
Walmart (WMT) Stock Price Under Pressure! Profit Misses Expectations for the First Time in Three Years, Soaring Tariff Costs Drag Down Profit Outlook
*August 22, 2025 - Transcript from the conference call: "Good morning/afternoon, everyone. Welcome to Walmart's Fiscal Year 2026 Second Quarter Earnings Conference Call. All participants will now be in a listen-only mode..."*

---

A web-based product fails a Single Historical Data Retrieval question in FINSEARCHCOMP due to the absence of search tool utilization.

**Question**
How much cash was used in investing activities by Apple in the 2021 fiscal year? (Answer in millions of dollars, rounded to nearest integer)

**Reasoning**
The user needs to know the amount of cash used in investing activities by Apple in the 2021 fiscal year, in millions of dollars, rounded to the nearest integer. First, I need to recall Apple's financial reports, specifically the cash flow statement for the 2021 fiscal year.
Apple's fiscal year typically ends in September, so the 2021 fiscal year would be the year ending September 2021. According to public financial data, in Apple's 2021 cash flow statement, the net cash flow from investing activities is usually negative because investment expenditures exceed income. The "cash used" here refers to the net amount of total cash outflows from investing activities minus inflows, but typically "cash used" in a question refers to the net cash outflow from investing activities.
Consulting Apple's 2021 10-K report, the investing activities section includes capital expenditures (like PP&E), acquisitions, investment securities, etc. **From memory**, was the net cash outflow from investing activities for Apple's 2021 fiscal year approximately 144,266 million dollars? Or is more accurate data needed.
Confirming again, in Apple's 2021 fiscal year cash flow statement, the net cash flow from investing activities was -144,266 million dollars. This means the cash used for investing activities was 144,266 million dollars. So the answer should be 144266.

**Response**
144266

**Search Sources**
**No search sources were used.**

---

A web-based product successfully solves a Complex Historical Investigation question by scoping its search to a reliable source (nasdaq.com) and then cross-verifying the retrieved data through reverse searches.

**Question**
What was the change in opening price compared to the previous closing price (in dollars, unadjusted) on the effective date of Apple Inc.'s (AAPL) most recent stock split? What was the split ratio? (Round to two decimal places)

**Reasoning**
...
Now, let's try searching NASDAQ for "AAPL closing price August 28 2020" to confirm the exact figure.

- **Searched for "AAPL closing price August 28 2020 site:nasdaq.com"**
  - *nasdaq.com* (x10)
- **Searched for "AAPL opening price August 31 2020 site:nasdaq.com"**
  - *nasdaq.com* (x10)

...
**Confirming closing price**
I'm searching for "AAPL August 28 2020 close 499.23" to verify the exact figure.

- **Searched for "AAPL August 28 2020 close 499.23"**

    - *finance.yahoo.com*
    - *statmuse.com*
    - *stockanalysis.com*
    - *investor.apple.com*
    - *financecharts.com*
    - *fool.com*

- **Searched for "AAPL August 31 2020 open 127.58"**

...

**Response**
The most recent stock split for Apple Inc. (AAPL) was effective on August 31, 2020, with a split ratio of 4-for-1.
On that date, the opening price was $127.58, compared to the previous closing price of $499.23 on August 28, 2020. The change was -$371.65.

