# OpenReview forum: "FinSearchComp: Towards a Realistic, Expert-Level Evaluation of Financial Search and Reasoning"
_ICLR.cc/2026/Conference — ICLR 2026 Poster_

### Official Review · Reviewer_i6JZ · 2025-10-27

**Soundness:** 3
**Presentation:** 3
**Contribution:** 3
**Rating:** 6
**Confidence:** 4

**Summary:**

FinSearchComp introduces the first open-source benchmark specifically designed to evaluate LLM agents in realistic financial search and reasoning scenarios. The benchmark replicates real financial analyst workflows through three task types: Time-Sensitive Data Fetching, Simple Historical Lookup, and Complex Historical Investigation. Developed with input from professional financial experts and a multi-stage quality assurance process, it contains 635 annotated questions across global and Greater China markets. Overall, FINSEARCHCOMP provides a rigorous, professional-grade testbed for assessing end-to-end financial search and reasoning in LLM-based agents.

**Strengths:**

1. Considering the critical importance of data timeliness in financial scenarios, the benchmark is designed from this perspective to test the capabilities of LLM agents.

2. The tasks are constructed through expert human validation rather than LLM-generated content, ensuring high-quality and reliability of the benchmark questions.

**Weaknesses:**

1. In the case study (page 25), one example shows a web-based LLM agent answering purely from memory, without using a search tool. In real scenarios, an LLM might appear confident while fabricating both reasoning and sources. Therefore, attributing errors solely to parametric memory is inaccurate (as noted in line 377), hallucination itself is another major factor contributing to model mistakes.

2. The paper contains a large amount of content, and to save space, the authors often refer readers to the appendix for details. This makes the reading experience somewhat incomplete. If possible, more of these details should be presented in the main text.

**Questions:**

1. Do the three task types mentioned in the paper (i.e., Time-Sensitive Data Fetching, Simple Historical Lookup, and Complex Historical Investigation) fully capture the main information retrieval scenarios encountered by financial analysts in real-world work? Are there any important task types missing, such as forecasting, report generation, or causal reasoning?

2. From Figure 3, models without search capabilities still achieve non-trivial scores on T2 and T3 questions. The paper attribute this to parametric memory from pre-training. However, could these results also indicate that many questions in T2 and T3 do not genuinely require search to answer, thus weakening their effectiveness in evaluating search-augmented reasoning?



#####

---

> ### Author Response · Authors · 2025-11-27
> **Response to Reviewer i6JZ — Part 1 of 2 (covers Q1\Q2)**
>
> ### Q1:
>
> Do the three task types mentioned in the paper (i.e., Time-Sensitive Data Fetching, Simple Historical Lookup, and Complex Historical Investigation) fully capture the main information retrieval scenarios encountered by financial analysts in real-world work? Are there any important task types missing, such as forecasting, report generation, or causal reasoning?
>
> ### A1:
>
> Thank you for the thoughtful question.
>
> For the retrieval layer that FINSEARCHCOMP is designed to evaluate, our answer is yes: the three task types (T1/T2/T3) capture the main information retrieval scenarios we observed in real-world analyst workflows. FINSEARCHCOMP was designed specifically as a benchmark for open-domain financial data search, and the scope is intentionally aligned with the forms of retrieval that professional analysts perform most frequently. Through our data-collection process involving 70 industry analysts (Section 2.4; Fig. 1), we repeatedly validated that the workflows captured by T1/T2/T3—time-sensitive data fetching, point-in-time historical lookup, and multi-period investigation—represent the core retrieval operations underpinning real-world financial analysis.
>
> Tasks such as forecasting, report generation, or causal reasoning are indeed important, but they belong to downstream reasoning rather than retrieval. These tasks depend on the ability to obtain accurate and timely data in the first place—the exact capability FINSEARCHCOMP aims to evaluate and the area where existing benchmarks are lacking. Our contribution is to fill this gap with a realistic, expert-curated, open-domain financial search benchmark; broader analytical tasks remain outside our intended scope.
>
> ### Q2:
>
> From Figure 3, models without search capabilities still achieve non-trivial scores on T2 and T3 questions. The paper attribute this to parametric memory from pre-training. However, could these results also indicate that many questions in T2 and T3 do not genuinely require search to answer, thus weakening their effectiveness in evaluating search-augmented reasoning?
>
> ### A2:
>
> Thanks for raising this point.
>
> We do not believe these results indicate that many T2/T3 questions do not genuinely require search. Rather, the non-zero scores produced by no-search models largely reflect parametric memory of widely available historical numbers, instead of showing that the tasks can be reliably solved without retrieval. In fact, our experiments show the opposite: FINSEARCHCOMP strongly differentiates systems with and without search.
>
> As reported in Figure 3, all no-search models score 0 on T1, confirming their inability to handle time-sensitive retrieval. Even for T2 and T3, enabling search yields substantial improvements (+40.8 for T1, +29.0 for T2, +8.1 for T3 on average), demonstrating that accurate performance requires accessing external sources rather than relying on memorized or approximate values. Moreover, a large fraction of T2/T3 questions involve fiscal-calendar alignment, multi-source validation, and long-horizon synthesis—tasks that cannot be handled reliably by parametric memory alone.
>
> Thus, these observations reinforce rather than weaken the benchmark’s purpose: FINSEARCHCOMP stresses search-augmented reasoning, and the results show that search tools make a decisive difference on T2/T3 as well as T1.

---

> > ### Author Response · Authors · 2025-11-27
> > **Response to Reviewer i6JZ — Part 2 of 2 (covers W1\W2)**
> >
> > ### Q3（W1）:
> >
> > In the case study (page 25), one example shows a web-based LLM agent answering purely from memory, without using a search tool. In real scenarios, an LLM might appear confident while fabricating both reasoning and sources. Therefore, attributing errors solely to parametric memory is inaccurate (as noted in line 377), hallucination itself is another major factor contributing to model mistakes.
> >
> > ### A3:
> >
> > Thank you for raising this point.
> >
> > We agree that hallucination and parametric memory both contribute to model errors. Our analysis in Section 4 was not intended to attribute mistakes to only one factor, but rather to illustrate a recurring pattern: when agents do not trigger search, they tend to rely on a mixture of partial parametric memory and hallucinated details, often producing confident but incorrect answers.
> >
> > This behavior highlights one of the core motivations for FINSEARCHCOMP. Real financial analysis requires verifying information against authoritative sources; relying solely on internal memory is inadequate. Our case study aims to show why robust search mechanisms are essential—not to pinpoint a single cause of error, but to illustrate the consequences of omitting retrieval in high-stakes domains. In the camera-ready version, we will revise the wording around line 377 to explicitly mention both parametric memory and hallucination, rather than attributing errors solely to parametric memory.
> >
> > ### Q4（W2）:
> >
> > W2: The paper contains a large amount of content, and to save space, the authors often refer readers to the appendix for details. This makes the reading experience somewhat incomplete. If possible, more of these details should be presented in the main text.
> >
> > ### A4:
> >
> > We appreciate the reviewer’s feedback.
> >
> > Because the benchmark involves substantial quality-control procedures, multi-stage expert validation, and detailed evaluation rubrics, we placed many large tables and rule lists in the appendix to stay within the page limit and maintain readability. That said, we agree that surfacing more of the most important details in the main paper would improve the reading experience.
> >
> > In the camera-ready version, we will integrate key elements—such as a concise summary of the core rubric rules, the essential components of the expert workflow, and a few representative examples—into the main text, while keeping full tables and rule lists in the appendix to preserve reproducibility.

---

> > > ### Comment · Reviewer_i6JZ · 2025-11-28
> > >
> > > Thanks for the response and the clarifications. Since this is a benchmark paper, I no longer have additional technical questions at this stage. For now, I lean to keeping my current score. Once the system enables editing score, I will reassess and consider whether a raise is appropriate.

---

### Official Review · Reviewer_UJjp · 2025-10-27

**Soundness:** 3
**Presentation:** 3
**Contribution:** 4
**Rating:** 6
**Confidence:** 5

**Summary:**

This paper introduces FinSearchComp, an open source benchmark for evaluating financial data search and reasoning via LLM agents. The benchmark covers 3 regimes of QA tasks: 1. Time sensitive data retrieval (real time quotes); 2. Simple historical lookups (point in time facts from records); 3. Complex historical investigations (multi-hop and multi step analysis across multiple sources, ex: search for best performing period). The authors also compared an extensive array of providers and models with regional origin clearly labeled. Further more, the analysis is ablated on greater China and western markets, with experiments faceted by region and provider. The dataset is 635 questions labeled by experts, and evaluations by LLM judges are grounded in human alignment of 97%.

**Strengths:**

1. Human experts involved in labeling process. Quality controls are included in curation.
2. Well motivated task breakdown into realtime, search, and aggregate query. Paper provides examples in appendix across different topics.
3. Coverage and experimental breakdown by regions and providers. 21 providers are experimented, and results highlight regional strengths. This allows for a fair and balanced evaluation of origin bias and current gaps in models for financial tool calling.
4. Open source commitment and good evaluation of tools. Ablations are conducted in api vs web enabled products.
5. Highlights clear SOTA gaps vs human annotations, gives a good future direction for development with the dataset (T3).
6. Handles realtime queries, not just point in time.

**Weaknesses:**

1. Limited scope to search insights, and the dataset is not focused on reasoning or future looking guidance. Most analysts are focused on providing guidance with research.
2. Limited 20 day window for search in August. Authors should consider adding different time seasons to encode seasonality in their analysis.
3. Limited details on data sourcing and licensing. How will the open source benchmark handle possible access issues in the underlying sources, if from different providers or services.
4. For T3, the scores are low but the paper doesn’t dive deeper into which sub-step fails. Is this a search issue, a tagging issue? Is this a reasoning failure by models? Context length issues?
5. Human labeling - details missing on how much time each labeled was given, what resources were available during the annotation process (terminals, APIs, fin service access)?

**Questions:**

1. For T1, T2, T3 can independent researchers fully reproduce the ground truth using the released artifacts and public sources? Is there any private information that requires subscriptions? Can you also specific what tools and time budget human labelers had during annotation?
2. Judge robustness: how did the judge alignment differ from the same model vs different families of models?
3. Time-stamping for T1: how are the authors handling live snapshots for reproducibility? Do authors encode universal time zones in their templates? How can this be reproduced for the original analysis in August vs today?
4. Could you quantify the error taxonomy in T3? Which error is most common? Misaligned fiscal calendars, units, etc?

---

> ### Author Response · Authors · 2025-11-27
> **Response to Reviewer UJjp — Part 1 of 2 (covers Q1\Q2\Q3)**
>
> ### Q1:
>
> For T1, T2, T3 can independent researchers fully reproduce the ground truth using the released artifacts and public sources? Is there any private information that requires subscriptions? Can you also specific what tools and time budget human labelers had during annotation?
>
> ### A1:
>
> All ground-truth answers in FinSearchComp are fully reproducible using publicly available data sources. **No private or subscription-only data is required to derive the ground truth.** While our expert annotators used professional financial tools (Wind, Tonghuashun iFinD, Factset, Bloomberg) to speed up lookup, these tools act only as *aggregators of public data*, not exclusive providers.
>
> For example, for the question: “During the period from January 1, 2023, to March 31, 2023, which constituent stock of the Nasdaq 100 had the highest average daily trading volume?”, annotators might consult Wind’s tables for convenience, but the underlying data is accessible from public sources such as SEC/EDGAR, stock exchange websites, or official market data portals.
>
> Annotators generally used Factset, Bloomberg, Wind, Tonghuashun iFinD, and sometimes directly used official websites (SEC, EDGAR, exchange websites, statistical bureaus).
>
> Time budget (based on our annotation process):
> - **T1 (Time-Sensitive Fetching):** <1–2 minutes/question
> - **T2 (Historical Lookup):** ~5–10 minutes/question
> - **T3 (Complex Investigation):** ~15–60 minutes/question (depending on complexity)
>
> Independent researchers can fully reproduce all labels using only public sources by following the released annotation templates and question definitions.
>
> ### Q2
>
> Judge robustness: how did the judge alignment differ from the same model vs different families of models?
>
>
> ### A2
>
> We ensured the robustness and impartiality of our evaluation primarily through the objective nature of the benchmark and a comprehensive validation of the judge's accuracy against human experts.
>
> 1. Objective Nature of Answers:
> FINSEARCHCOMP questions demand objective, verifiable financial data points, leading to answers that are highly converged and deterministic (with defined tolerance bands). This intrinsic objectivity ensures that the LLM-as-a-Judge cannot exhibit preferential scoring toward any specific model or model family, as the "correctness" is fact-based and narrowly defined.
>
> 2. Judge Model and Self-Scoring Check:
> Our LLM-as-a-Judge system utilized a robust model (DouBao-0115). We specifically checked the performance of the judging model itself when acting as a candidate model and found **no evidence of inflated or preferential scoring**. The model's evaluation remains anchored to the strict rubrics, irrespective of the candidate being scored.
>
> 3. Cross-Task and Cross-Regional Alignment:
> We validated the judge's alignment by comparing its scoring results against human-verified labels across all tasks and both subsets. The alignment accuracy remained consistently high across all dimensions, confirming the reliability of the 0-1 scheme:
>
> The consistent accuracy across varying complexity levels (T1 to T3) and different languages confirms the judge's reliable performance irrespective of the source or family of the candidate model being evaluated.
>
> ### Q3:
> 1. Time-stamping for T1: how are the authors handling live snapshots for reproducibility?
> 2. Do authors encode universal time zones in their templates?
> 3.  How can this be reproduced for the original analysis in August vs today?
>
> ### A3:
>
> Thank you for raising this important point. We clarify the reproducibility notion that applies to time-sensitive search tasks like T1 (similar to benchmarks such as BrowseComp):
>
> 1. **Exact environment-level reproduction (e.g., “the web state in August”) is infeasible** for any open-web retrieval benchmark, because search indexes, API responses, and website contents evolve continuously. This limitation is structural to all real-world search tasks.
>
> 2. What we guarantee is **protocol-level reproducibility**, which is the accepted standard: FINSEARCHCOMP provides fixed templates, explicit timestamps, consistent tool interfaces, post-market-close evaluation timing, and a judge rubric, ensuring fairness and comparability within each evaluation round.
>
> 3. **Time specifications in our templates are explicitly encoded and normalized (e.g., using consistent time-zone interpretation)** so that tools can reliably interpret the intended time window.
>
> Re-running the benchmark today will naturally yield different raw search results compared to August, but the evaluation **procedure** is reproducible, stable, and fully specified—aligned with prior open-domain agent benchmarks and designed for scientific rigor in time-sensitive financial retrieval tasks.

---

> > ### Author Response · Authors · 2025-11-27
> > **Response to Reviewer UJjp — Part 1 of 2 (covers Q4)**
> >
> > ### Q4:
> >
> > Could you quantify the error taxonomy in T3? Which error is most common? Misaligned fiscal calendars, units, etc?
> >
> > ### A4:
> >
> > We first clarify why T3 does not admit a fully quantified and mutually exclusive error taxonomy.
> > **T3 requires multi-hop retrieval, multi-period alignment, and heterogeneous data fusion, so errors typically arise from interacting retrieval + reasoning steps rather than isolated categories.** Different agents follow different trajectories, making the error surface unstable for precise percentage breakdowns.
> >
> > That said, our manual inspection of representative T3 samples (Section 4) identifies **three dominant high-level error families** that consistently explain model failures:
> >
> > 1. **Insufficient or stale retrieval (most common).**
> >    Agents often retrieve incomplete evidence, use outdated data, or fail to issue deeper follow-up queries. This matches the observations in Fig. 3 and Sections 4.1–4.4.
> >
> > 2. **Temporal or calendar misalignment.**
> >    Fiscal-year vs calendar-year mismatches, quarter-boundary shifts, or misinterpreted reporting windows, also discussed in our ambiguity-mitigation guidelines (Table 4).
> >
> > 3. **Unit / normalization inconsistencies.**
> >    Currency conversion errors, thousand/million scaling mistakes, or unadjusted corporate actions, which become more pronounced when synthesizing multiple periods.
> >
> > Because T3 solutions are multi-step and model-dependent, these error families frequently interact; therefore, forcing a rigid quantitative taxonomy would be less informative than recognizing these dominant patterns. Our analysis aims to surface these representative failure modes to guide future improvements in agent reasoning and retrieval.

---

### Official Review · Reviewer_f1kM · 2025-10-31

**Soundness:** 2
**Presentation:** 3
**Contribution:** 2
**Rating:** 4
**Confidence:** 3

**Summary:**

This paper introduces FINSEARCHCOMP, an open-source benchmark designed to evaluate the search and reasoning capabilities of LLM-based agents in the financial domain.  FINSEARCHCOMP comprises 635 expert-curated questions divided into three realistic tasks. The dataset spans Global and Greater China markets and was developed through a rigorous quality-control pipeline involving 70 financial experts. The authors evaluate 21 models , finding that Grok 4 (web) and DouBao (web) lead the Global and Greater China subsets, respectively , though all models significantly trail human expert performance.

**Strengths:**

1. The benchmark's construction was overseen by 70 professional financial experts (50 annotators, 20 senior reviewers) from prominent institutions, ensuring high domain relevance and realism

2. The paper provides a broad evaluation of the current landscape by testing a large set of 21 models, which includes 12 web-based products and 9 APIs.

**Weaknesses:**

1. The primary novelty, financial searching evaluation, is a superficial extension that does not meaningfully advance methodologies for financial retrieval or agent benchmarking. The dataset is modest in scale and task complexity remains shallow, yielding only incremental progress beyond prior financial QA benchmarks.

2. Although the work claims to evaluate agentic capability, it does not assess actual agent frameworks or contrast general-purpose agents with financial-domain agents. Relying solely on API vs. web endpoints provides no evidence that the benchmark measures planning, tool choice, error recovery, or other core agent behaviors.

3. The exclusive use of binary 0/1 accuracy collapses retrieval, reasoning, source selection, and tool-use behavior into a single outcome, preventing attribution of failure sources and limiting diagnostic interpretability.

4. Multisource retrieval is presented as essential to financial tasks, yet the benchmark includes no metrics for source coverage, retrieval depth, or authority resolution. Thus, it remains unclear whether models perform multisource search at all, weakening a central premise of the benchmark.

5. The annotation process depends on senior expert arbitration but reports neither annotator disagreement nor arbitration rates, leaving uncertainty about reliability, transparency, and potential bias in conflict resolution.

6. Market coverage is overstated: the benchmark focuses mainly on Western and Greater China markets and omits major regions (e.g., Japan, India, emerging economies). Coupled with the limited dataset size, this constrains representativeness and community impact.

7. Although real-time correctness is highlighted, the protocol lacks detail on how temporal consistency is enforced between model inference and ground-truth snapshots, especially for volatile assets, creating concerns about reproducibility.

**Questions:**

See weakness above

---

> ### Author Response · Authors · 2025-11-27
>
> Thank you for your suggestions, and we address your concerns as follows.
>
> ### Q1: Novelty & Task Complexity of FinSearchComp
> We respectfully disagree that our FinSearchComp lacks novelty or the tasks are shallow, and we would like to clarify our novelty as follows:
> - **Real-World Relevance**: Our three task types (T1-T3) directly mirror professional analyst workflows with escalating complexity. For T1 (Time-Sensitive Data Fetching), analysts typically spend around 1-2 minutes per query for quick references. This task is also commonly performed by non-expert investors in scenarios such as checking the current price of their investments. For T2 (Simple Historical Lookup) represents the most frequent task type, with individual analysts performing 10-30 such queries daily for financial analysis, peer comparisons, and modeling. Each lookup averages 5-10
> minutes, accounting for data validation. For T3 (Complex Historical Investigation) demands the highest time investment per-query, often requiring 15-60 minutes for comprehensive data retrieval and calculation, based on the complexity of the data retrieval and calculation steps. While less frequent (fewer than 20 queries per analyst monthly), these investigations
> and calculations are important building blocks in financial analysis and reports.
> - **Task Complexity**: As demonstrated in the paper, the tasks are professional and complex. We define three task types aligned with daily analyst workflows. Each requires at least one external tool call, and each question has a single, fully objective answer. These tasks test core search & reasoning skills. The models need to search from the open web environment, compare different information sources, and reason with professional finance knowledge.
> - **Expert-Level Quality Assurance**: We engaged 70 professional financial experts (50 annotators + 20 senior reviewers) holding advanced degrees from institutions, ensuring questions reflect genuine professional standards, not synthetic academic exercises. The entire expert cohort includes practitioners from prestigious institutions such as Citadel, J.P. Morgan, Deutsche Bank, Nomura, Citigroup, Credit Suisse, Harvest Fund, Zhongou Asset Management, CITIC Securities, and Huatai Securities.
> - **Industry Recognition**: Leading model developers—including MiniMax and Kimi—have adopted FinSearchComp for their latest model releases (**MiniMax-M2 and Kimi-K2-thinking**) in the last two months, validating the practical utility for measuring real-world capabilities.
>
>
> ### Q2: The Scale of FinSearchComp
> We would like to emphasize that the scale of FinSearchComp is well-suited for its purpose as a professional, expert-level benchmark.
> * With 635 expert-curated questions, FinSearchComp surpasses several other agent benchmarks in scale. For instance, the VALS Finance Agent Benchmark includes only 50 questions, while the Terminal Bench contains 80. Our benchmark's current size is already highly effective at distinguishing model capabilities, as demonstrated in reports from **MiniMax-M2** and **Kimi-K2-thinking**. While we can expand further if necessary, the primary limitation is not the number of questions, but the quality of expert validation.
> * Additionally, increasing the scale would incur significantly higher time and financial costs for testing. Given these considerations, we believe that the current scale of 635 questions is both appropriate and efficient for the goals of FinSearchComp.
>
>
> ### Q3: Web endpoints do not represent agentic capabilities.
> We would like to clarify that all web-based products are **exactly agentic systems**, representing optimized solutions developed by each foundation model company. These systems integrate the base LLM with customized tools and appropriate system prompts to deliver the best user experience. Evaluating these end-to-end products provides a more accurate and fair assessment than testing individual models with a single standardized framework, which may not equally suit all models due to differences in tool-calling formats and training data.
> - Our analysis of Grok 4's web product reveals that its system prompt indicates the presence of over 10 specialized tools (e.g., code execution, web search, page browsing, multiple X search tools, PDF browsing, etc.). Successful use of the system requires: (i) selecting the appropriate tools, (ii) formulating effective queries, and (iii) synthesizing multi-step information.
> - As detailed in Section 4.4 and Appendix D, we document systematic failures such as insufficient search depth, failure to activate plugins, and retrieval of outdated data—issues that are typically expected when agent planning or error recovery fails. Our Complex Historical Investigation tasks (T3) explicitly require the model to recognize insufficient initial searches and reformulate queries accordingly.

---

> > ### Author Response · Authors · 2025-11-27
> >
> > ### Q4: Failure mode analysis.
> > While we use binary scoring for standardization, **in Section 4 and Appendix D**, we conducted extensive trace analysis to diagnose failure sources across tasks:
> > - **T1 (Time-Sensitive)**: For products without financial APIs, >70% of errors stem from retrieving stale data (intraday or previous-day prices). For products with financial plugins, >70% of failures occur when models bypass the plugin and retrieve unreliable web data from static pages.
> > - **T2 (Historical Lookup)**: ~40% of errors arise from sourcing news sites instead of authoritative filings. Additionally, some models rely on parametric memory without search, leading to frequent inaccuracies.
> > - **T3 (Complex Investigation)**: Models with financial APIs correctly invoke them in <30% of cases, often defaulting to web search or querying databases incorrectly. High-performing models consistently verify data accuracy through cross-referencing authoritative sources.
> >
> > These findings (detailed in Section 4 and Appendix D with concrete examples) demonstrate that our evaluation framework enables clear attribution of failure modes—retrieval depth, source selection, and tool-use behavior—despite binary scoring. We've added explicit failure taxonomy in the revised manuscript to enhance diagnostic transparency.
> >
> > ### Q5: The benchmark includes no metrics for source coverage, retrieval depth, or authority resolution.
> > FinSearchComp evaluates end-to-end task completion using realistic, expert-designed questions that closely mirror actual analyst workflows. As outlined in Sections 2.2 and 2.3, our 70 professional financial experts created tasks based on their daily practices, with the core objective being to obtain the correct answer—not to document the search process. In real-world financial analysis, the primary criteria for success are the accuracy and reliability of the results, not the intermediate retrieval steps.
> >
> > The need for multisource retrieval is inherent in our task design. For example, in Complex Historical Investigation (T3), questions explicitly require synthesizing data across multiple time periods, sources, and reporting formats—tasks that cannot be solved without effective multisource search (Table 2, Section 2.2). The substantial performance gap between T1 and T3 (Figure 5), along with case study findings in Section 4.4 indicating that "few successful attempts are confined to queries requiring fewer than five data points," empirically demonstrate that models must engage in deeper, multisource retrieval to succeed in more complex tasks.
> >
> > While we recognize the importance of analyzing search sources, the design of various LLM web products limits our ability to fully document their search sources.
> >
> > ### Q6: Quality control.
> > We would like to emphasize the quality control process of FinSearchComp, as well as the annotator accuracy. Our primary focus was on process design to ensure the definitive ground truth, reducing reliance on raw agreement rates alone.
> >   1. Arbitration Process Rationale: Our quality control (Section 2.4) uses senior experts not just to judge competence, but to review the entire, independently generated trace of the two junior annotators. By examining the methodology and data sources used by each expert, the senior expert can decisively identify which answer and process is correct, ensuring the resulting ground truth is robust, rather than a mere consensus vote.
> >   2. Initial Annotation Quality: For the historical tasks requiring manual expert lookup, the initial agreement among our annotators was strong:
> >     - T2 (Simple Historical Lookup): Initial Annotator Accuracy was 89.8%.
> >     - T3 (Complex Historical Investigation): Initial Annotator Accuracy was 80.0%.
> >     - (Note: T1 ground truth is generated via API calls from external databases, not subjective annotation.)
> >   3. Final Quality Assurance Loop: We maintained an iterative feedback loop with our senior experts during arbitration. This process continued until a 30% random sample of arbitrated questions revealed no factual errors, with only minor wording adjustments sometimes required. This final check provides strong evidence of the quality and convergence achieved through the arbitration process.

---

> > > ### Author Response · Authors · 2025-11-27
> > >
> > > ### Q7: Market coverage.
> > > We would like to emphasize that the US and Greater China markets are the two largest markets. We understand that global financial markets are highly diverse, however, due to the high costs of finding corresponding experts, certain major regions, such as Japan, India, and emerging economies, are currently beyond the scope of FinSearchComp.
> > >   1. **Prioritized Focus**: Our development process required extensive resources, specifically highly proficient financial experts, which resulted in a high cost per question. We strategically prioritized the Global (Western) and Greater China markets to first validate the methodology in the most frequently traded and high-value domains for our sponsoring experts. Expanding coverage to include additional major regions is a crucial goal for our future work.
> > >   2. **Core Value is Methodology**: We assert that the core value of FinSearchComp lies in its methodology —— the design of three expert-level tasks (T1, T2, T3) and the end-to-end evaluation framework that assesses tool orchestration, reasoning, and time-sensitivity. The conclusions drawn from this benchmark regarding failure modes (e.g., stale data, shallow search, unit normalization failures) and the proven necessity of specialized tool use are universally applicable and can be extended to evaluate financial agents operating in any regional market. The benchmark serves as a foundational proof-of-concept for realistic financial agent evaluation.
> > >
> > >
> > > ### Q8: Details on how temporal consistency is enforced between model inference and ground-truth snapshots.
> > >   We enforce temporal consistency through a rigorous, time-synchronized protocol and permissive rubrics for volatile assets.
> > >   1. **Protocol and Ground Truth**: As stated in our methodology in section 2.6, we conduct all T1 evaluations after the official market close (local market time) on each evaluation day. This standardizes the retrieval time for all models to the latest closing data. The ground truth is generated dynamically by API calls executed at this standardized time. The evaluation code for this process, which defines the dynamic ground truth, is part of the open-source evaluation harness.
> > >   2. **Robustness Checks**: To account for volatility and minor source discrepancies, we established differentiated rubrics. For highly volatile assets like FX rates, the valid range is defined as the day's high-low range augmented by a buffer. The reliability of this process is confirmed by the high accuracy of the LLM-as-a-Judge on T1: **91.5% (Global) and 91.7% (Greater China)**.
> > >   3. **Reproducibility**: The entire evaluation harness, including the time-sensitive API calling scripts, is released open-source for full reproducibility.

---

### Official Review · Reviewer_9TaR · 2025-11-01

**Soundness:** 4
**Presentation:** 4
**Contribution:** 3
**Rating:** 8
**Confidence:** 3

**Summary:**

This paper presents a benchmark for evaluating LLMs on real-world financial reasoning tasks that involve real-time data retrieval, historical lookup, and multi-source analysis. It includes 635 expert-verified queries, split into 337 Global and 298 Greater China tasks, created and reviewed by 70 finance professionals. The authors evaluate 21 models and show that even top web-enabled systems still fall short of human analysts, especially on time-sensitive or multi-hop reasoning. The main value lies in its holistic setup, as it tests the complete agent process from search and tool use to reasoning and final judgment within realistic financial workflows rather than isolated question answering.

**Strengths:**

The useful part of this work is that it evaluates the whole agent process in a financial setting where the answer can actually change over time. The model is expected to go out to a source (web or a financial plugin), pull time aligned data, match it to the task (for example the right fiscal date or the right currency), and then produce an answer that is checked by a rubric. It covers three analyst style tasks: time sensitive queries, point in time historical queries, and longer multi period questions, so it is not only checking whether the model can retrieve a document.

The task split into time sensitive, point in time historical, and longer multi period questions is sensible and shows where models start to fail. They validated the LLM as judge setup against human labels and got about 95 percent agreement, so the scoring looks trustworthy. They also released the scoring rules and judge prompts, which makes the benchmark easier to reproduce.

**Weaknesses:**

A limitation of the work is the effective scale of the benchmark. Although the dataset contains 635 tasks in total, the division into two regions and three task types results in relatively small groups; most subsets contain fewer than 150 items, and the most demanding setting (T3) has only about 80 examples. This makes fine-grained comparisons across models, topics, or markets less statistically robust than the headline number suggests. The coverage is also concentrated on U.S. and Greater China markets, with limited representation of other regions, so the current release should be viewed as financially realistic but not yet globally comprehensive.

The paper is reasonably transparent in terms of evaluation methodology, since the scoring rules and judge prompts are provided. However, reproducibility of the exact numbers remains difficult, because the time-sensitive tasks rely on live market data and some of the evaluated systems are proprietary web products.

**Questions:**

1. Was the same LLM judge and prompt used for the global/English and Greater China/Chinese parts, and did you check whether the 0–1 scheme behaves the same across languages or for partly correct long answers?

2. Performance appears to depend on the region of both the model and its attached tools, with international systems performing better on global tasks and Chinese systems doing better on the Greater China subset. How much of these differences reflect model reasoning capability versus disparities in access to region-specific data sources or plugins?

3. Given that 635 expert tasks is already expensive, is there a plan for semi-automatic growth, and how do you see this benchmark sitting next to Finance Agent Benchmark or FinEval?

---

> ### Author Response · Authors · 2025-11-27
> **Response to Reviewer 9TaR — Part 1 of 2 (covers Q1 and Q2)**
>
> ### Q1:
>
> Was the same LLM judge and prompt used for the global/English and Greater China/Chinese parts, and did you check whether the 0–1 scheme behaves the same across languages or for partly correct long answers?
>
> ### A1:
>
> Thank you for this crucial question regarding the consistency and reliability of our evaluation protocol across different linguistic and task settings.
>
> 1.  Consistency of LLM-as-a-Judge Protocol:
> We confirm that the same judge model (identical model checkpoint and configuration) was used for both the Global/English and Greater China/Chinese subsets. The core evaluation function that maps answers to a binary 0–1 score is therefore fully shared across all subsets.
>
>
> 2.  Language-Specific Prompts:
> While the underlying scoring methodology is identical, we used English judge prompts for Global tasks and Chinese judge prompts for Greater China tasks. The prompt structure, scoring rubric, and decision rules remain the same; only the language is changed to ensure correct interpretation of detailed scoring criteria (e.g., tolerances, units, definitional standards).
>
> 3.  Verification of 0-1 Scheme Across Subsets and Tasks:
> We empirically validated the robustness of the 0–1 scoring scheme through human evaluation by comparing LLM-as-a-Judge outputs against human-verified labels on a representative sample from both subsets. The results are summarized below:
>
>     | Task | Subset | Accuracy (%) |
>     | :--- | :--- | :--- |
>     | T1 | Global | 91.5 |
>     | T1 | Greater China | 91.7 |
>     | T2 | Global | 96.8 |
>     | T2 | Greater China | 95.5 |
>     | T3 | Global | 97.4 |
>     | T3 | Greater China | 99.8 |
>
> To directly address the question about partially correct long answers, the judge protocol for T2 and T3 explicitly requires that all key points (e.g., required dates and associated values) must be present and correct for the answer to receive a score of 1. This rule is applied identically in both languages. During human validation, we also manually inspected cases where answers contained only a subset of required information. In both the English and Chinese settings, the judge consistently assigned a score of 0 in such cases, showing that the binary scheme behaves uniformly across languages and does not reward partial completeness even for long, multi-step answers.
>
>
> ### Q2:
> Performance appears to depend on the region of both the model and its attached tools, with international systems performing better on global tasks and Chinese systems doing better on the Greater China subset. How much of these differences reflect model reasoning capability versus disparities in access to region-specific data sources or plugins?
>
> ### A2:
> Thank you for this important question. Our analysis indicates that the regional performance differences arise from both (i)
>
> ### 1. Evidence from End-to-End Products (Model + Native Tools)
> We first examine full products where models use their native ecosystems of tools/plugins. In this setting, the divergence is large: U.S. products perform better on the Global subset, while Chinese products perform better on the Greater China subset.
>
> | Task | U.S. Model (Product) Ratio | Chinese Model (Product) Ratio |
> | :--- | :--- | :--- |
> | T2 (Simple Historical Lookup) | 102% | 50% |
> | T3 (Complex Historical Investigation) | 244% | 136% |
>
> These results suggest that access to proprietary, region-specific data sources or plugins plays a substantial role.
> ### 2. Evidence from API Models with a Unified, Fixed Tool
> To isolate model-internal capability, we conducted an ablation study: raw API models were all attached to the *same fixed tool set*. This greatly reduces tool-access differences.
>
> | Task | U.S. Model (API + Fixed Tool) Ratio | Chinese Model (API + Fixed Tool) Ratio |
> | :--- | :--- | :--- |
> | T2 | 89% | 67% |
> | T3 | 137% | 106% |
>
> The gap narrows compared to the end-to-end setting, but does not disappear. This demonstrates that intrinsic model reasoning ability and alignment with region-specific corpora contribute to regional performance bias.
>
>
> ### **Conclusion**
> Because performance differences persist even when tool access is controlled, we conclude that **both intrinsic reasoning capability and tool/data disparities** jointly shape the observed regional skew. The much larger divergence in the end-to-end system comparison (e.g., 244% vs. 136% on T3) indicates that **region-specific tools and data integration amplify these differences**.

---

> > ### Author Response · Authors · 2025-11-27
> > **Response to Reviewer 9TaR — Part 2 of 2 (covers Q3 and additional clarifications)**
> >
> > ### Q3:
> >
> > Given that 635 expert tasks is already expensive, is there a plan for semi-automatic growth, and how do you see this benchmark sitting next to Finance Agent Benchmark or FinEval?
> >
> > ### A3:
> >
> > ### 1. Plan for Semi-Automatic Growth
> > Yes. Given the high cost of expert curation, we have already designed a semi-automatic expansion pipeline. The workflow consists of:
> >
> > - **Source Data Ingestion:** Collecting authoritative content from official filings and regulated public disclosures.
> > - **Automated Draft Generation:** Using LLMs to generate candidate questions and definitive answers grounded in the extracted official documents.
> > - **Rule-Based Quality Filtering:** Applying strict heuristic filters to ensure factual alignment, unit and definition consistency, and adherence to our scoring constraints.
> >
> > This approach enables efficient scaling: we have generated ~1,500 additional questions with only five person-days of effort.
> >
> > ### 2. Positioning Relative to Existing Benchmarks
> >
> > #### Finance Agent Benchmark
> > FINSEARCHCOMP provides a broader and more demanding evaluation:
> > - **Time-Sensitive Retrieval:** We include T1 (real-time quotes and market data), which FAB does not assess.
> > - **Scope and Difficulty:** Our T3 (Complex Historical Investigation) requires multi-period synthesis, corporate action adjustments, and unit normalization—capabilities beyond FAB’s static-data search.
> > - **Openness:** FINSEARCHCOMP is fully open-source and provides 635 real-world expert tasks.
> >
> > #### FinEval
> > FinEval evaluates financial-domain knowledge primarily through classification, extraction, and generation:
> > - **Context Provided by Default:** Tasks usually give the relevant data directly to the model.
> > - **Different Objective:** FINSEARCHCOMP uniquely evaluates *end-to-end, open-domain search + reasoning*, not just knowledge recall or NLU/NLG.
> >
> > ### 3. Impact and Adoption
> > FINSEARCHCOMP has already been referenced in technical reports from new model releases such as Minimax and Kimi, indicating recognition by the research community as a rigorous and meaningful benchmark for financial search agents.

---

> > > ### Comment · Reviewer_9TaR · 2025-11-27
> > >
> > > Thanks for the further details. I will remain my rating.

---

### Meta-Review · Area_Chair_7B5h · 2026-01-08

**Summary:**

This paper presents FinSearchComp, a valuable benchark for evaluating LLM-based agents on realistic finncial search and reasoning tasks. Reviewers broadly upport the work's contribution and quality, recognizng the rigorous expert curation process involving 70financial professionals from leading institutions, cmprehensive evaluation across 21 models, and the wel-motivated three-task structure (T1: Time-Sensitive ata Fetching, T2: Simple Historical Lookup, T3: Compex Historical Investigation).

Key concerns raised iclude: (1) Limited dataset scale—while 635 expert-anotated questions is substantial, subdivision into tw regions and three task types results in relatively mall subsets (<150 items), limiting statistical robutness for fine-grained comparisons; (2) Geographic sope—coverage focuses on Western and Greater China makets, with limited representation of other major regons (Japan, India, emerging markets); (3) Experimentl design—evaluation relies on web-based products andAPIs without detailed analysis of which failures ste from model reasoning vs. tool access/data disparitis, and the binary 0/1 scoring lacks granularity for iagnosing specific failure modes (e.g., retrieval deth, source selection, tool-use behavior); (4) Clarit and presentation—important methodological details ae relegated to appendices, affecting readability; (5 Scope limitations—the benchmark evaluates search an retrieval, not broader analytical tasks (forecastin, report generation, causal reasoning) that completefinancial analysis workflows.

Despite these limitatons, reviewers acknowledge the benchmark's significat methodological contributions: realistic task desig grounded in professional analyst workflows, rigorou multi-stage quality assurance, comprehensive evaluaion framework, and openness via public release with eproducible evaluation harness. The paper demonstrats clear SOTA gaps against human expert performance ad highlights actionable failure modes (stale data, sallow search, temporal misalignment, unit normalizaton failures).

**Reviewer Concerns:**

Addressed Concerns:
- The authors provided detailedresponses to all reviewer questions, demonstrating teir engagement with the feedback
- Clarified the LLMas-a-Judge protocol consistency across languages, wih empirical validation showing 91.5-99.8% accuracy aross task types and subsets
- Explained regional perormance differences through controlled ablations, shwing that both intrinsic model reasoning capability nd tool/data disparities jointly shape outcomes
- Oulined a concrete semi-automatic expansion pipeline wth ~1,500 additional questions already generated
- Psitioned FinSearchComp relative to existing benchmars (Finance Agent Benchmark, FinEval), emphasizing it focus on end-to-end open-domain search + reasoning
 Provided extensive failure mode analysis in Section4 and Appendix D, identifying root causes despite biary scoring (stale data >70%, plugin bypass, insuffiient search depth, temporal misalignment, unit normaization failures)
- Clarified that all 21 evaluated web-based products" are agentic systems with specialzed tools, not simple API calls
- Documented the quaity control process with strong initial annotator agreement (89.8% for T2 80.0% for T3) and iterative arbitration with seniorexperts

Outstanding Concerns:
- Dataset scale remais limited for fine-grained regional/topic-specific cmparisons—while 635 questions is substantial, subdivsion into two regions and three task types yields susets <150 items, constraining statistical robustness- Geographic coverage focused on Western and GreaterChina markets; expansion to Japan, India, and emergig economies remains future work, limiting global repesentativeness
- Binary 0/1 scoring, while defended hrough detailed failure analysis, lacks fine-graineddiagnostics for source coverage, retrieval depth, an authority resolution that could provide additional nsights
- Important methodological details remain inappendices, potentially affecting discoverability an reproducibility for researchers
- Scope intentionaly limited to search/retrieval evaluation, not broade analytical tasks (forecasting, report generation, cusal reasoning); however, this is acknowledged as inentional design rather than a limitation

Overall Asessment:
The authors addressed substantive concerns hrough detailed empirical evidence, clarifications o methodology, and documentation of failure modes. Th work makes meaningful contributions to financial agnt evaluation despite the noted limitations on scale scope, and geographic breadth.

**Reviewer Scores:**

Reviewer 9TaR (Original Score: 8 - Accept, good papr poster):
Would likely MAINTAIN score of 8. The revewer's concerns regarding scale limitations and geogaphic scope remain valid, but the authors' detailed esponses on LLM-as-a-Judge consistency, failure modeanalysis, and methodology clarifications demonstratestrong engagement. The benchmark's value for financil agent evaluation is well-justified through empiricl evidence and industry adoption (MiniMax, Kimi). Th responses do not fundamentally alter the core assesment but reinforce the paper's methodological rigor.
Reviewer f1kM (Original Score: 4 - Marginally belowacceptance threshold):
Would likely INCREASE to 5-6.The reviewer raised concerns about novelty, task comlexity, and evaluation methodology. The authors' detiled responses (Q1-Q3) addressing benchmark novelty rounded in professional analyst workflows, the scalejustification (635 questions surpasses FAB's 50), an clarification that web-based products are agentic sstems with 10+ tools substantially address the core oncerns. The extensive failure mode analysis (Sectio 4, Appendix D) and industry adoption demonstrate th benchmark's practical value. The rebuttal evidence uggests the paper merits reconsideration as marginaly acceptable.

Reviewer UJjp (Original Score: 6 - Maginally above acceptance threshold):
Would likely MANTAIN or INCREASE to 6-7. The reviewer's concerns abut data sourcing, human labeling process, and T3 errr taxonomy were thoughtfully addressed. The authors rovided concrete documentation of quality control (8.8% for T2, 80.0% for T3 initial accuracy), detailedthe time budgets for annotation, and confirmed full eproducibility using public sources. The error taxonmy clarification (insufficient/stale retrieval, tempral misalignment, unit normalization) provides suffiient diagnostic insight despite binary scoring. Highconfidence in the evaluation methodology is now morejustified.

Reviewer i6JZ (Original Score: 6 - Margially above acceptance threshold):
Would likely MAINTIN score of 6. The reviewer's primary concerns—whethr the three task types fully capture financial analyt workflows and whether T2/T3 questions genuinely reuire search—were clearly addressed. The authors demostrated through Figure 3 that search enables substanial improvements (+40.8 for T1, +29.0 for T2, +8.1 fr T3) and explained that non-zero no-search scores rflect parametric memory, not task feasibility withou retrieval. The revision incorporating more details n the main text also addresses the presentation concrn about content relegated to appendices.

---

### Decision · Program_Chairs · 2026-01-26

Accept (Poster)